# Potent and protective IGHV3-53/3-66 public antibodies and their shared escape mutant on the spike of SARS-CoV-2

Qi Zhang[1,11], Bin Ju[2,11], Jiwan Ge [3,11], Jasper Fuk-Woo Chan [4,5,11], Lin Cheng[2,11], Ruoke Wang [1,11], Weijin Huang [6,11], Mengqi Fang[1], Peng Chen [1], Bing Zhou[2], Shuo Song[2], Sisi Shan [1], Baohua Yan[3], Senyan Zhang [3], Xiangyang Ge[2], Jiazhen Yu[2], Juanjuan Zhao[2,7], Haiyan Wang[2], Li Liu [5,8], Qining Lv[1], Lili Fu[1], Xuanling Shi[1], Kwok Yung Yuen [4,5], Lei Liu[2], Youchun Wang [6✉], Zhiwei Chen [4,5,8✉], Linqi Zhang [1,9,10✉], Xinquan Wang [3✉] & Zheng Zhang [2,7✉]

Neutralizing antibodies (nAbs) to SARS-CoV-2 hold powerful potentials for clinical interventions against COVID-19 disease. However, their common genetic and biologic features remain elusive. Here we interrogate a total of 165 antibodies from eight COVID-19 patients, and find that potent nAbs from different patients have disproportionally high representation of IGHV3-53/3-66 usage, and therefore termed as public antibodies. Crystal structural comparison of these antibodies reveals they share similar angle of approach to RBD, overlap in buried surface and binding residues on RBD, and have substantial spatial clash with receptor angiotensin-converting enzyme-2 (ACE2) in binding to RBD. Site-directed mutagenesis confirms these common binding features although some minor differences are found. One representative antibody, P5A-3C8, demonstrates extraordinarily protective efficacy in a golden Syrian hamster model against SARS-CoV-2 infection. However, virus escape analysis identifies a single natural mutation in RBD, namely K417N found in B.1.351 variant from South Africa, abolished the neutralizing activity of these public antibodies. The discovery of public antibodies and shared escape mutation highlight the intricate relationship between antibody response and SARS-CoV-2, and provide critical reference for the development of antibody and vaccine strategies to overcome the antigenic variation of SARS-CoV-2.

A full list of author affiliations appears at the end of the paper.

The rapid international transmission and emergence of SARS-CoV-2 variants poses a serious challenge to global health, and may render our current antibody and vaccine strategies ineffective. Like other coronaviruses, SARS-CoV-2 utilizes the receptor-binding domain (RBD) of envelope homo-trimeric spike (S) glycoprotein to interact with cellular receptor angiotensin-converting enzyme-2[1–7]. Binding with ACE2 triggers a cell membrane fusion cascade for viral entry. Neutralizing antibodies (nAbs) that effectively block RBD-ACE2 interaction represent potential prophylactic and therapeutic options and could also guide vaccine design.

We and other have isolated a large number of nAbs against SARS-CoV-2[3,8–26] and provided important insights into the antibody response during natural SARS-CoV-2 infection. Regardless of isolation methods used, majority of the potent nAbs target the RBD, with only small representatives recognize the other regions such as the N-terminal domain, S2, or quaternary structures on the trimeric spike glycoprotein[14,15,17,22,23]. RBD-targeting nAbs appear to be more viral species-specific, while those against S2 region tend to be more cross-reactive with other human and animal coronaviruses[8–10,27–29]. These cross-reactive antibodies are in general less neutralizing and recognize more conserved epitopes outside the RBD[15,17]. While uncertain how these antibodies work in vivo, it is conceivable that they work in concert to bind and inhibit viral entry. Given the complex genetic background and sophisticated antibody mutation and maturation processes in vivo, it is expected that antibody response elicited by SARS-CoV-2 infection varied among different individuals. However, recent studies have identified pattern of convergence in antibody lineages cross different COVID-19 patients[8,18,21]. Particularly, a disproportionally higher number of nAbs was derived from IGHV3-53/3-66 family (differed in only one amino acid in framework region 1)[3,11,25], suggesting that different individuals could share some public patterns and pathways in antibody response during SARS-CoV-2 infection. Interestingly, similar findings have also been reported for individuals infected by dengue[30] and HIV-1[31], after influenza vaccination[32], and in other immune settings[33–35].

Here, we interrogated a total of 165 antibodies isolated from eight infected COVID-19 patients through our earlier studies[19]. Crystal structures and mutagenesis analysis revealed the existence of public antibodies shared among three of the eight patients. These antibodies shared IGHV3-53/3-66 in their heavy chain and characterized by their high neutralizing potency to SARS-CoV-2, similar angle of approaching to RBD, and competitive capacity and binding footprints with ACE2 on RBD. One of their representative, P5A-3C8, demonstrated highly protective efficacy in a golden Syrian hamster model against SARS-CoV-2 infection. However, single mutation at position 417 emerged during in vitro selection was found to confer resistance to these antibodies. Mutations at position 417 have recently been identified in SA501Y.V2 (B.1.351) variant from South Africa and in BR501Y. V3 (P.1) from Brazil and found capable of substantially reducing antibody and vaccine efficacy including those already approved for emergence use[28,36–40]. We have therefore investigated the impact of all naturally occurring mutations at position 417 on binding and neutralization of public antibodies. Taken together, the insights on these public antibodies will improve our understanding on the common features in antibody responses among different individuals and, most importantly, to capitalize on these features for antibody drug and vaccine development to overcome continued emergence of SARS-CoV-2 variants.

## Results

### Potent neutralizing antibodies prefer IGHV3-53/3-66 usage.

We previously reported the isolation and characterization of 206 RBD-specific monoclonal antibodies (mAbs) with 165 distinct sequences derived from single B cell of eight SARS-CoV-2-infected individuals[19]. From the initial studies of 18 antibodies, we identified a few antibodies with potent anti-SARS-CoV-2 neutralization activity that correlates with their competitive capacity with ACE2 for RBD binding. Here, we further characterize the remaining 147 antibodies for their binding and neutralizing activities. Using pseudovirus for initial screening, we identified 13 nAbs with half-maximal inhibitory concentrations ($IC_{50}$) ranging from 0.0014 to 0.0996 µg/mL (9.33 to 664.00 pM) (Supplementary Fig. 1a, Table 1). The $IC_{50}$ of remaining antibodies, however, spans between 0.1 and 50 µg/mL or higher (666.67 and 333.33 nM) (Supplementary Table 1). The top 13 nAbs also demonstrated strong inhibitory activity against live SARS-CoV-2 based on focus reduction neutralization tests (FRNT) (Supplementary Fig. 1b)[19]. For instance, the $IC_{50}$ for the best antibody P5A-1B9 reached as low as 0.0043 µg/mL (28.67 pM) and the $IC_{80}$ 0.0441 µg/mL (294.00 pM), at least tenfold more potent than our previously reported ones[19]. These nAbs demonstrated high yet varying binding affinity to the SARS-CoV-2 RBD measured by surface plasmon resonance (SPR) (Supplementary Fig. 1c and Table 1). All except P5A-2G9, P5A-1D2, and P2B-1A10, displayed single digit nanomolar binding affinity. Apart from P5A-3B4, these nAbs shared strong competitive capacity with ACE2 in binding to SARS-CoV-2 RBD, suggesting their potential mechanism of neutralization (Supplementary Fig. 1d and Table 1). Most interestingly, of the top 13 nAbs, 7 were found to use IGHV3-53/3-66 and paired predominantly with IGKV1-9*01 (Table 1). Four of the seven were derived from P#5 (P5A-1D1, P5A-1B8, P5A-1D2, and P5A-3C8), whereas two from P#2 (P2C-1F11 and P2B-1A10) and one from P#22 (P22A-1D1) (Fig. 1a, b, Table 1). Such high prevalence (53.8%) and diverse origin among the top neutralizers indicated that IGHV3-53/3-66 represented one major and public antibody responses against SARS-CoV-2. Furthermore, their CDR3 length varied from 9 to 15, located in the shorter range among the total 165 RBD-specific antibodies identified (Fig. 1c). Their somatic hypermutation (SHM) were generally low and some reached 0% for heavy chain (P22A-1D1) or light chain (P5A-1B8 and P2C-1F11) (Table 1, Fig. 1d), suggesting their potency does not require extensive maturation process. Recent reports have also recognized disproportionally high prevalence of IGHV3-53/3-66 with limited somatic mutations among SARS-CoV-2 patients[11,25].

### The public antibodies resemble ACE2 in binding epitope on SARS-CoV-2 RBD.

To reveal the structural basis for potent neutralization of the public antibodies, we determined crystal structures of P22A-1D1 (2.40 Å), P5A-3C8 (2.36 Å), and P5A-1D2 (2.60 Å) complexed with SARS-CoV-2 RBD (Fig. 2a and Supplementary Table 2). We previously determined the complex structure of another public antibody P2C-1F11 with RBD at a resolution of 2.96 Å (PDB ID: 7CDI) and used it here for head to head comparison[41]. As shown in Fig. 2a, the four public antibodies (P22A-1D1, P5A-1D2, P5A-3C8, and P2C-1F11) bound to the RBD with a nearly identical angle of approach. The estimated clash volume with ACE2 was about ~20,000 Å$^3$ (Fig. 2a), consistent with biochemical data showing strong capacities to compete with ACE2 for binding to SARS-CoV-2 RBD (Table 1). Their heavy chains share similar buried surfaces on the RBD. The estimated areas are 729 Å$^2$ for P22A-1D1, 725 Å$^2$ for P5A-3C8, 840 Å$^2$ for P5A-1D2, and 775 Å$^2$ for P2C-1F11 (Fig. 2b and Supplementary Table 3). In contrast, the buried surface areas of the light chain are rather different. P22A-1D1 (409 Å$^2$) and P5A-3C8 (548 Å$^2$) are significantly larger than P5A-1D2 (164 Å$^2$) and P2C-1F11 (204 Å$^2$) (Fig. 2b and Supplementary Table 3).

**Table 1 Binding capacity, neutralizing activity, and gene family analysis of 13 monoclonal Abs isolated from Patient #5, Patient #2, and Patient #22.**

| Patient | mAbs | Binding to RBD | | Pseudovirus (μg/ml) | | Live virus (μg/ml) | | Heavy chain | | | | Kappa chain (K) or Lambda chain (L) | | | |
| | | Kd (nM) | Competing w/ ACE2 | $IC_{50}$ | $IC_{80}$ | $IC_{50}$ | $IC_{80}$ | IGHV | HCDR3 | HCDR3 length | SHM (%) | IGK(L)V | K(L)CDR3 | K(L) CDR3 length | SHM (%) |
|---|---|---|---|---|---|---|---|---|---|---|---|---|---|---|---|
| P#5 | P5A-1B9 | 3.41 | +++ | 0.0014 | 0.0053 | 0.0043 | 0.0441 | 4-59*01 | ASNGQYYDILTGQPPDYWYFDL | 22 | 0.70 | K4-1*01 | QQYYSTPLT | 9 | 0.00 |
| P#22 | P22A-1D1 | 5.79 | +++ | 0.0038 | 0.0625 | 0.0198 | 0.1321 | 3-53*01 | ARDRDYYGMDV | 11 | 0.00 | K1-9*01 | LHLNSYRT | 8 | 0.38 |
| P#5 | P5A-2G7 | 3.95 | +++ | 0.0044 | 0.0287 | 0.1814 | 0.8355 | 4-61*01 | ARERCYYGSGRAPRCVWFDP | 20 | 0.34 | L2-14*01 | SSYTSSTLVV | 11 | 0.74 |
| P#5 | P5A-1D1 | 6.83 | +++ | 0.0096 | 0.0691 | 0.0189 | 0.0743 | 3-53*01 | ARDLYYYGMDV | 11 | 0.35 | K1-9*01 | QQLNSYPT | 8 | 0.76 |
| P#5 | P5A-1B8 | 4.28 | +++ | 0.0115 | 0.0501 | 0.0168 | 0.0857 | 3-53*01 | ARETLAFDY | 9 | 1.40 | K1-9*01 | QQLNSYPPA | 9 | 0.00 |
| P#5 | P5A-2G9 | 15.94 | +++ | 0.0158 | 0.1466 | 0.0113 | 0.1187 | 3-33*01,3-33*06 | ARWFHTGGYFDY | 12 | 0.00 | L5-37*01 | MIWPSNALYV | 10 | 0.35 |
| P#5 | P5A-1D2 | 14.02 | +++ | 0.0186 | 0.1025 | 0.0273 | 0.4325 | 3-53*01 | ARALQVGATSDYFDY | 15 | 1.40 | L1-40*01 | QSCDSSLSVVV | 11 | 1.11 |
| P#5 | P5A-3C8 | 1.30 | +++ | 0.0206 | 0.1031 | 0.0112 | 0.1499 | 3-53*01 | ARDLQEHGMDV | 11 | 1.05 | K1-9*01 | QHLNSYPPGYT | 11 | 1.14 |
| P#2 | P2C-1F11* | 3.64 | +++ | 0.0286 | 0.1195 | 0.0323 | 0.1779 | 3-66*01,3-66*04 | ARDLVVYGMDV | 11 | 1.75 | K3-20*01 | QQYGSSPT | 8 | 0.00 |
| P#2 | P2B-2F6* | 5.57 | +++ | 0.0500 | 0.6074 | 0.4074 | 2.4309 | 4-38-2*02 | ARAVVGIVVPAAGRRAFDI | 20 | 0.69 | L2-8*01 | SSYAGSNNLV | 10 | 0.00 |
| P#2 | P2B-1A10 | 38.41 | +++ | 0.0974 | 0.7446 | 0.0639 | 0.3053 | 3-53*01 | AREGPKSITGTAFDI | 15 | 0.35 | K1-33*01, K1D-33*01 | QQYDNLPMYT | 10 | 0.38 |
| P#5 | P5A-3B4 | 1.16 | + | 0.0993 | 1.0657 | 0.0561 | 1.0080 | 5-51*01 | ARRDSTYGGNTDY | 13 | 0.35 | L1-44*01 | AAWDDSLNGVV | 11 | 0.00 |
| P#5 | P5A-3C12 | 8.47 | +++ | 0.0996 | 0.4679 | 0.2636 | 2.6783 | 2-5*02 | AHSLFLTVGYSSWSPFDY | 19 | 0.00 | K4-1*01 | QQYSTPHT | 9 | 0.00 |

The program IMGT/V-QUEST was applied to analyze gene germline, complementarity determining region (CDR) 3 length, and somatic hypermutation (SHM). The CDR3 length was calculated from amino acids sequences. The SHM frequency was calculated from the mutated nucleotides. Antibody binding to RBD was presented either by $K_D$ or by competing with ACE2 where "+++" indicates >80% competition and "+" 20–50%. $IC_{50}$ represents the half-maximal whereas $IC_{80}$ the 80% inhibitory concentrations in the pseudovirus and live SARS-CoV-2 neutralization assay.
*Published in the reference (Ju, et al. [19]).

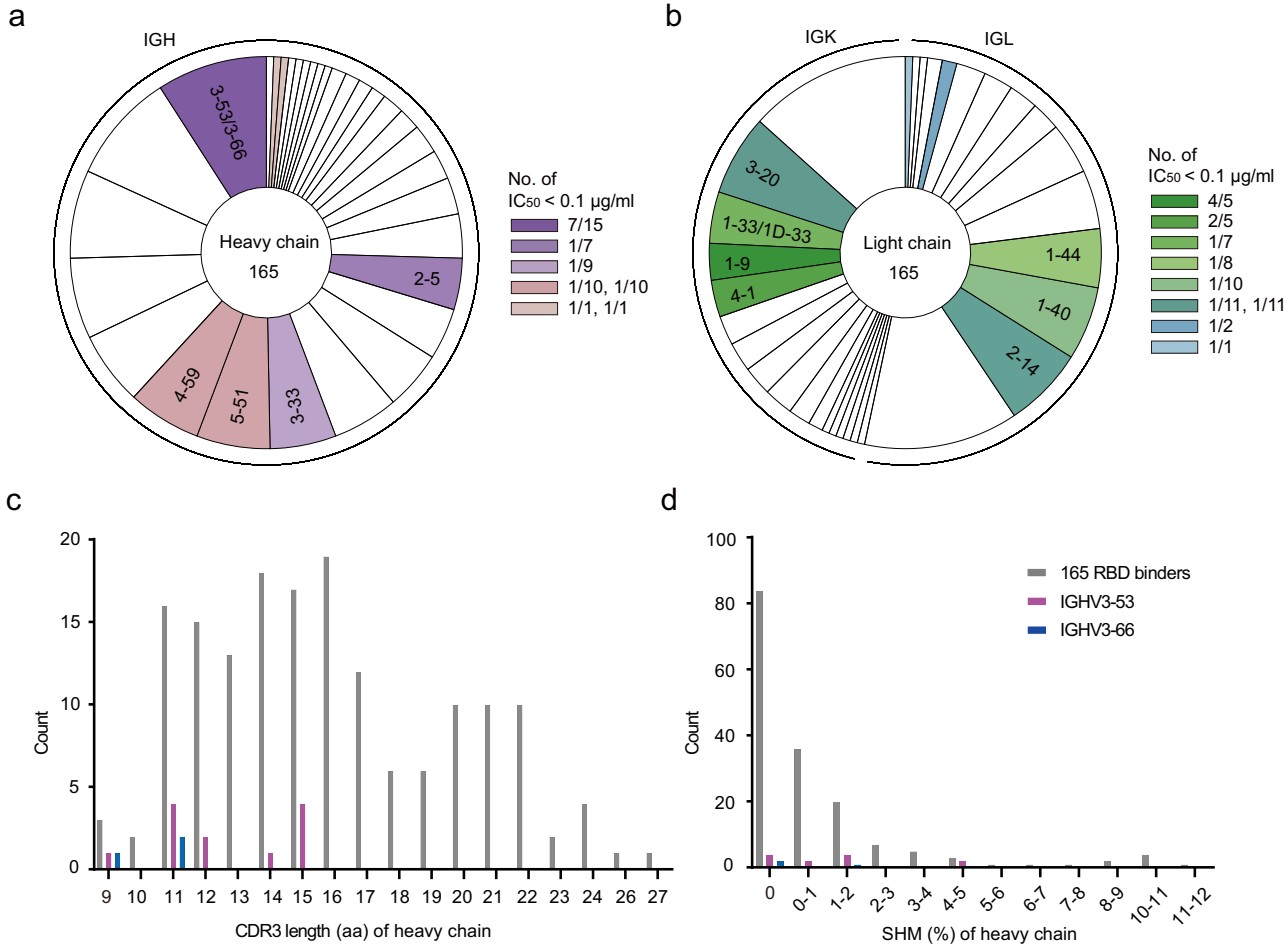

**Fig. 1 Preferred usage of IGHV3-53/3-66 among the potent neutralizing antibodies. a**, **b** Lineage analysis for heavy and light chains in pie charts. The numbers in the center represent the number of RBD-specific antibodies. Each slice represents a unique clone and proportional to its own size. **c**, **d** Counts of various CDR3 length and somatic hypermutation from IGHV3-53 and IGHV3-66 as well as RBD binders. Source data are provided as a Source Data file.

Interestingly, P5A-3C8, P22A-1D1, B38, and CC12.1 use IGKV1-9 as the light chains and engage a larger binding interface than those of IGKV3-20, including P2C-1F11, CC12.3, and CV30 (Supplementary Table 3)[3,25,42]. The larger buried areas are translated into more epitope residues. For instance, P22A-1D1 and P5A-3C8 have 28 and 31 epitope residues on the RBD, whereas P5A-1D2 and P2C-1F11 have 22 and 23, respectively (Supplementary Table 4). Furthermore, the epitopes of these public antibodies significantly overlap with the ACE2-binding residues on RBD. Out of 17 ACE2-binding residues on RBD, P22A-1D1 shared by 15, P5A-3C8 by 16, P5A-1D2 by 10, and P2C-1F11 by 11 (Fig. 2c). The similar angles of approach to and the large overlap in binding residues on the RBD suggest that the binding mode of these four public antibodies resemble that of ACE2 to SARS-CoV-2.

**Shared binding interface among the public antibodies**. The IGHV3-53 and IGHV3-66 share the identical germline amino acid sequence except one residue. It is therefore expected that the four public antibodies shared their binding features to RBD primarily through residues in the heavy chain. As shown in Fig. 3a, all three HCDRs are involved in the binding of these four public antibodies to the RBD. Heavy chain sequence alignments showed that the HCDR1 and HCDR2 are highly conserved, whereas the HCDR3 are rather different (Supplementary Fig. 2). Of note, P5A-1D2 has a longer HCDR3 (15 residues) than the rest three antibodies (11 residues). In the shared HCDR1-RBD interface, the conserved

HCDR1 residues G26, F27, T28/I28, S31, N32, and Y33 interact with RBD residues L455, K458, Y473, A475, G476, S477, and N487. In the shared HCDR2-RBD interface, interactions are largely mediated through HCDR2 residues Y52, S53, G54, S56 and Y58 and RBD residues T415, G416, K417, D420, Y421, K458 and N460. One unique feature shared by the public antibodies is the participation of three conserved tyrosines (Y33, Y52, and Y58) in forming a network of hydrophobic and hydrophilic interactions with the RBD (Fig. 3b). For example, the Y33 forms extensive hydrophobic interactions with RBD K417, Y421, L455 and F456 (Fig. 3b). Its side chain –OH also forms a conserved hydrogen bond with the main chain oxygen atom of RBD L455 (Fig. 3b). Another unique and shared feature is the interactions mediated by the -SGGS- segment in the HCDR2. Apart from the close contacts through Van der Waals forces, specific hydrogen-bonding interactions also occur between the beginning S53 and ending S56 with RBD Y421 and D420, respectively (Fig. 3c). In addition, RBD Y421 also forms a conserved hydrogen bond with main chain N atom of the G54 (Fig. 3c).

**Minor yet discernable differences among the public antibodies**. Despite of common and shared features, the public antibodies also demonstrated some minor differences due to their sequence and structure variations. P22A-1D1, P5A-3C8, and P2C-1F11 have the same 11-residue long HCDR3, but actual sequence varies. For example, the -RDYYG- in P22A-1D1 is replaced by -LQEHG- in P5A-3C8 and by -LVVYG- in P2C-1F11 (Table 1).

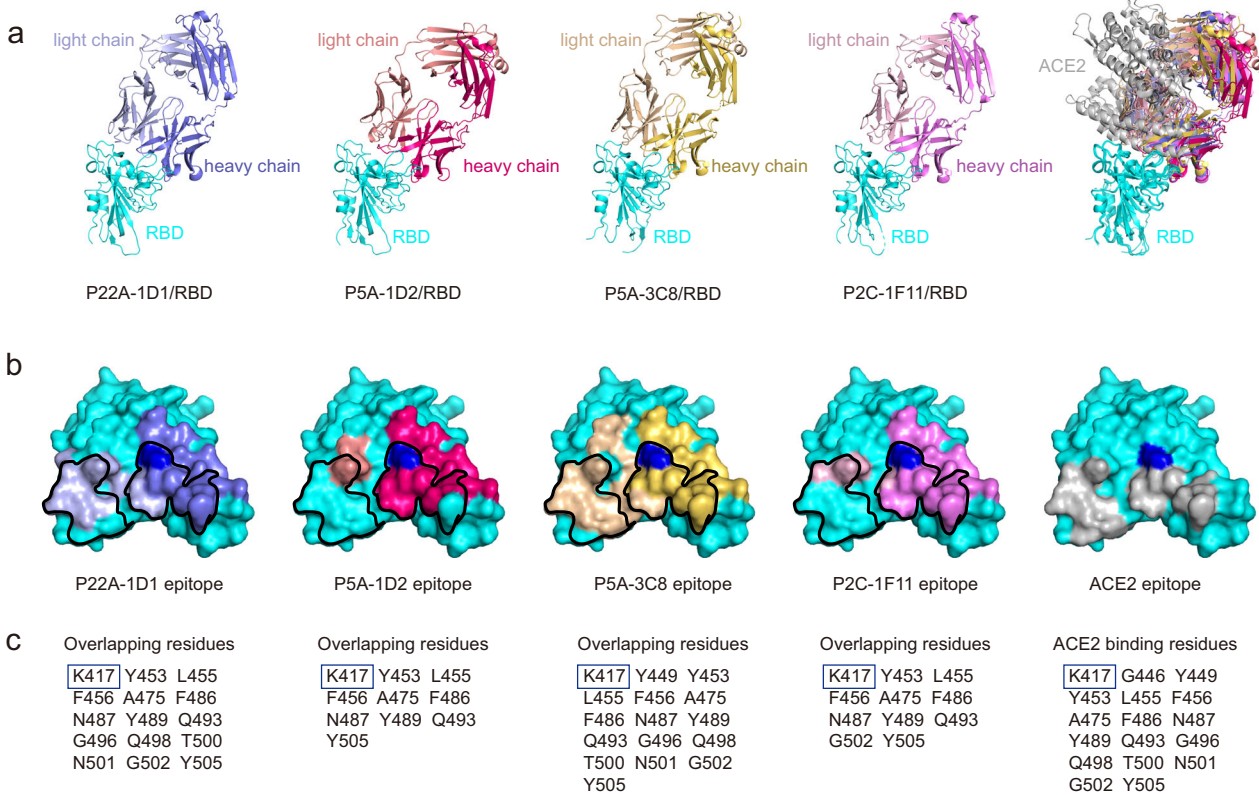

**Fig. 2 Structure and binding features of public antibodies to SARS-CoV-2 RBD. a** The crystal structures of RBD and Fab complexes. RBD was colored in cyan. P22A-1D1 Fab was colored in slate and light blue, P5A-1D2 in hot pink and salmon, P5A-3C8 in yellow orange and light yellow, and P2C-1F11 (PDB ID: 7CDI) in violet and pink, for heavy chain and light chain, respectively. **b** The footprint of Fabs and ACE2 on SARS-CoV-2 RBD. The color of the epitope was depicted as in **a**. The epitope of ACE2 was colored by black. K417 was highlighted with blue. **c** Binding residues shared with ACE2 are listed. K417 was highlighted with blue box.

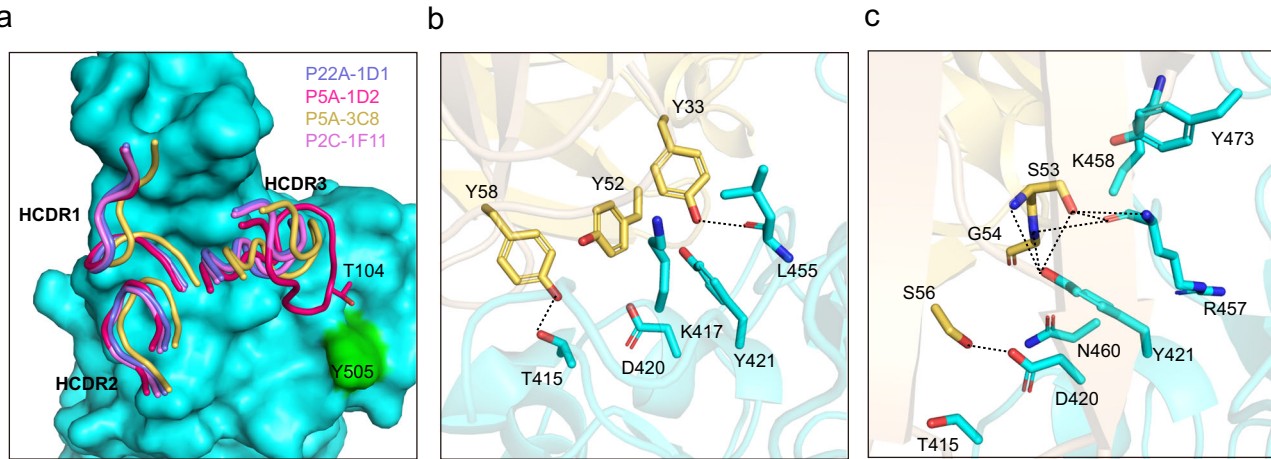

**Fig. 3 Shared binding interface among the public antibodies. a** Conserved HCDR1, HCDR2, and different HCDR3. RBD was colored in cyan and shown as surface. CDR loops of the heavy chain were shown as ribbon with the same color in Fig. 2. **b** The interactions between the three conserved tyrosine at HCDR1 and HCDR2. **c** The interactions between HCDR2 -SGGS- segment, and RBD. Hydrogen bonds were shown as black dashed line and P5A-3C8/RBD complex was used as an example in **b**, **c**.

Therefore, although interacting with the same RBD residues such as F456, N487, Y489 and Q493, the specific residues in the HCDR3 in mediating such interactions are different. Compared to the other three, P5A-1D2 has a relatively longer HCDR3 with 15 residues (Fig. 3a and Table 1), providing more residues to interact with RBD. For instance, the T104 at the tip of the P5A-1D2 HCDR3 has interactions with RBD Y505, which is absent in

other three HCDR3-RBD interfaces (Fig. 3a). RBD Y505 is instead recognized by the light chain of P22A-1D1, P5A-3C8 and P2C-1F11, and appears to serves as an anchor residue for light chain binding (Supplementary Fig. 3). However, recognition by the long HCDR3 of P5A-1D2 resulted in significant change in the side chain conformation of Y505, precluding Y505 serving as an anchor for P5A-1D2 light chain binding (Supplementary Fig. 3).

To further dissect the impact of epitope residues on the binding of public antibodies, we conducted single-site alanine scanning mutagenesis for the 15 epitope residues shared among the public antibodies. All mutant spikes were successfully expressed on the surface of HEK 293T cells, as measured by the median fluorescence intensity of the control S2 antibody through flow cytometry. However, of the 15 mutant residues, 12 resulted in more than 80% reduction in the binding of the four public antibodies although some antibodies are more sensitive than others (Supplementary Fig. 4, highlighted in gray boxes). For example, Y421A and F456A have broad impact on all four public antibodies, whereas T415A, Y473A, and N487A on three of the four. On the other hand, K417A, D420A, L455A, R457A, N460A, and Y489A only reduced binding for two of the four antibodies. Y505A appears to have more profound impact on P22A-1D1 than the rest three antibodies. Lastly, 9 out of the 15 mutant residues also resulted in significant reduction of ACE2 binding to the surface expressed spike glycoprotein. These residues are highlighted in orange boxes including T415A, Y421A, L455A, F456A, R457A, Y473A, N487A, Y489A, and Y505A. The shared impact of these residues on ACE2 and the public antibodies support the abovementioned structural analysis (Fig. 2). Taken together, these results indicate that some minor differences do exist among the four public antibodies despite of their overall similarity, which may account for their minor differences in binding and neutralizing activities.

**Rapid emergence of resistant mutants under the selective pressure of public antibody in vitro**. To explore the potential emergence of SARS-CoV-2 resistant strains to public antibodies, we selected escape mutants by growing a replication-competent recombinant VSV-SARS-CoV-2 in the presence of P22A-1D1 or P5A-3C8. The mutant viruses emerged quickly during 4 days culture in Passage 1 (P1), at the concentration of 6, 250-fold and 2, 500-fold higher in IC$_{50}$ for P22A-1D1 and P5A-3C8, respectively. The resistant strains from P1 were further selected in Passage 2 (P2) for additional 4 days with increasing concentrations of P22A-1D1 or P5A-3C8. Sequence analysis of the resistant strains from the highest antibody concentration identified K417N and K417T mutations in P22A-1D1-treated wells and K417T mutation in P5A-3C8-treated wells, all of which were present in 20 out of the 20 clones analyzed. Of note, K417N/T mutation has been recently identified in the SA501Y.V2 (B.1.351) variant from South Africa and in BR501Y.V3 (P.1) from Brazil and found capable of substantially reducing antibody and vaccine efficacy including those already approved for emergence use[28,36–40]. Through analysis of a total of 1,234,882 unique genome sequences in GISAID database (by May 17th, 2021), we found that 5,992 sequences bearing the K417N mutation, 11,734 sequences containing K417T mutation, 20 sequences with K417R mutation, and 3 sequences with K417E mutation (https://cov.lanl.gov/content/index, [43]), indicating residue at 417 is highly variable and likely under strong immune selection. We then produced mutant RBDs containing each of the K417R/A/E/N/T and tested their binding affinities to public antibodies and ACE2. Except that of P2C-1F11 where 4.4-fold increase was found, binding to mutant K417R RBD among the rest public antibodies was similar to that of WT, likely due to the similar positive-charged side chains between residue K and R (Fig. 4a, c). In contrast, K417A/E/N/T mutants substantially reduced binding by P22A-1D1 and P5A-1D2, while impact on P5A-3C8 and P2C-1F11 was relatively moderate (Fig. 4a, c).

We then constructed pseudovirus bearing each of the K417R, K417A, K417E, K417N, or K417T mutation and analyzed their neutralizing sensitivity to the public antibodies. Consistent with

binding analysis, P22A-1D1, P5A-3C8 and P2C-1F11 remain sensitive to K417R pseudovirus, except P5A-1D2 had neutralizing activity below the detection limit (BDL) even when tested at the highest concentration (1 μg/mL) (Fig. 4b, d). K417A/E/N/T mutants resulted in complete resistance to P5A-3C8, P22A-1D1, and P5A-1D2 while remaining sensitive to P2C-1F11 (Fig. 4b, d). To dissect out the potential molecular interactions accounting for the resistance, we summarized and compared the interactions between public antibodies and SARS-CoV-2 K417 residue within 4 Å cutoff in Supplementary Table 5. Salt bridges were found highly involved in the binding of K417 residue with P22A-1D1, P5A-1D2, and P5A-3C8, but not with P2C-1F11. K417A/E/N/T mutations disrupted these salt bridges, leading to loss of neutralizing activity. This result highlights the critical role of this particular residue on mediating binding and neutralization among 3 out of 4 public antibodies (Supplementary Table 5). One possible way to mitigate the emergence and spread of resistant strains is to use combination approach to maintain antibody potency as already been reported[13].

**Protection of hamsters from SARS-CoV-2 infection by public antibody**. To assess the public antibody's protection efficacy in vivo, we tested P5A-3C8 in a golden Syrian hamster model of SARS-CoV-2 infection. The hamsters were given an intraperitoneal injection of P5A-3C8 at a dose of 5 mg/kg, or same dose of VRC01, an anti-HIV-1 antibody as negative control. Twenty-four hours later, hamsters were challenged intranasally with $1 \times 10^3$ plaque-forming units (PFU) of SARS-CoV-2 HK13 strain and monitor for body weight, and virological and immunohistochemical changes in our days after virus challenge. Compared to the VRC01 control group, the total viral loads in the P5A-3C8 group were significantly reduced in lung, demonstrated by both viral RNA (genomic RNA and subgenomic RNA) and infectious virus titer measured by PFU (Fig. 5a–c). The P5A-3C8 group experienced a slight weight loss during the first 2 days after virus challenge and began to recover to gain body weight afterwards while those in the control group continued to drop without recovery (Fig. 5d). Furthermore, immunofluorescence (IF) staining of viral NP antigen only detected sporadic positive cells in the lung sections of P5A-3C8 treated animals. In contrast, diffuse NP expression was present in large areas of alveoli in VRC01 treated group. (Fig. 5e). This result shows P5A-3C8 confers strong protection in vivo against live SARS-CoV-2 infection.

**Discussion**

It is largely unknown whether different individuals use similar antibody genes in response to SARS-CoV-2 infection. With a few findings on the convergent antibody responses to SARS-CoV-2 spike antigens[8,18], there has been little structural evidence for their similarity[11,25]. Here, we defined the public antibody responses from multiple SARS-CoV-2-infected individuals and revealed their structural features at atomic levels. Our findings have provided more insights into the common features in antibody response during natural infection, and to facilitate antibody drug and vaccine development against SARS-CoV-2 infection.

The identification of public antibody in SARS-CoV-2 infection is intriguing, given the potential mutants of SARS-CoV-2 occurring among infected individuals during the spread. However, our findings are consistent with recent discoveries where convergent clonotypes with highly similar sequences were shared across different COVID-19 patients[8,18]. Various studies appear to identify different heavy chain gene clonotypes such as VH3-30, VH3-33, VH1-24, and VH1-58[8,16,18]. Our study has extended the list of public antibody clonotypes to VH3-53 and VH3-66

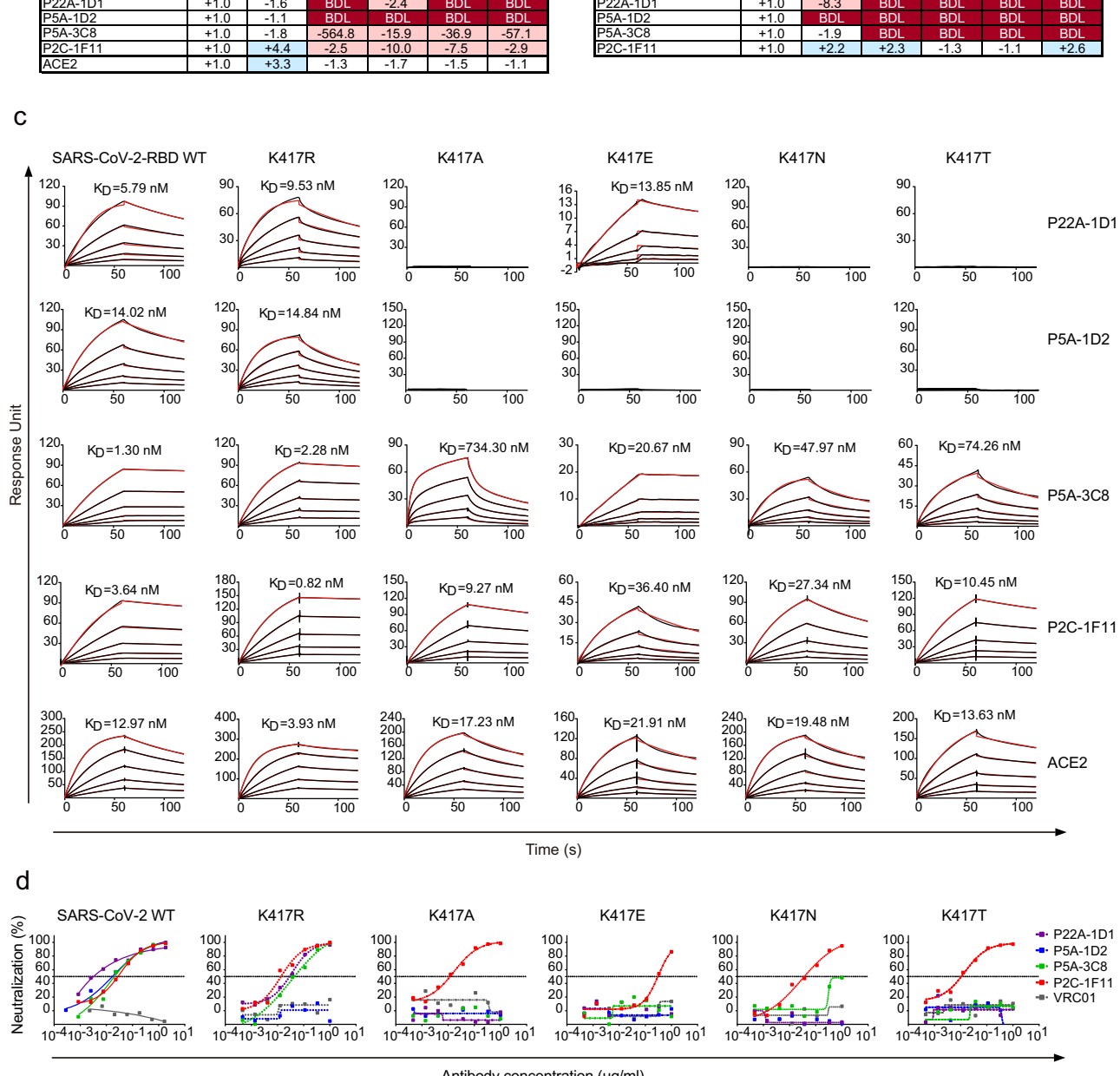

**Fig. 4 Susceptibility of SARS-CoV-2 K417 variants to binding and neutralization of public antibodies.** Values indicate the fold changes in **a** binding affinity ($K_D$) and **b** half-maximal inhibitory concentrations ($IC_{50}$). The symbol "-" indicates increased resistance and "+" increased sensitivity. Those $K_D$ or $IC_{50}$ values highlighted in red, resistance increased at least twofold; in blue, sensitivity increased at least twofold; and in white, resistance or sensitivity increased less than twofold. BDL (Below Detection Limit) indicates the highest concentration of mAbs failed to bind or reach 50% neutralization. **c** The individual antibodies were captured on protein A covalently immobilized onto a CM5 sensor chip followed by injection of purified soluble SARS-CoV-2 WT and K417/R/A/E/N/T mutant RBDs at five different concentrations. The black lines indicate the experimentally derived curves while the red lines represent fitted curves based on the experimental data. **d** Comparison of public antibodies' neutralization against pseudovirus bearing WT, K417R, K417A, K417E, K417N, or K417T mutant SARS-CoV-2 spike on the pseudovirus. VRC01 is an HIV-1 specific antibody used here as a negative control. Data shown are from at least two independent experiments. Source data are provided as a Source Data file.

although both of which have been reported for a few human-derived nAbs against SARS-CoV-2[3,12,14–16,21,22,26]. These analyses suggest high prevalence of public nAbs among infected patients which may represent one of most important attributes of neutralizing activities in infected humans.

Importantly, we provided the structural basis for the common features among these public antibodies. We found that the footprint of public antibodies on RBD and the angle of approaching to RBD are rather similar to ACE2, providing structural explanation for their strong capacity in competing with ACE2 in binding to RBD. However, minor differences exist among these public antibodies. This is largely reflected in their binding difference to a panel of mutant spike glycoprotein expressed on the cell surface. Importantly, virus escape studies

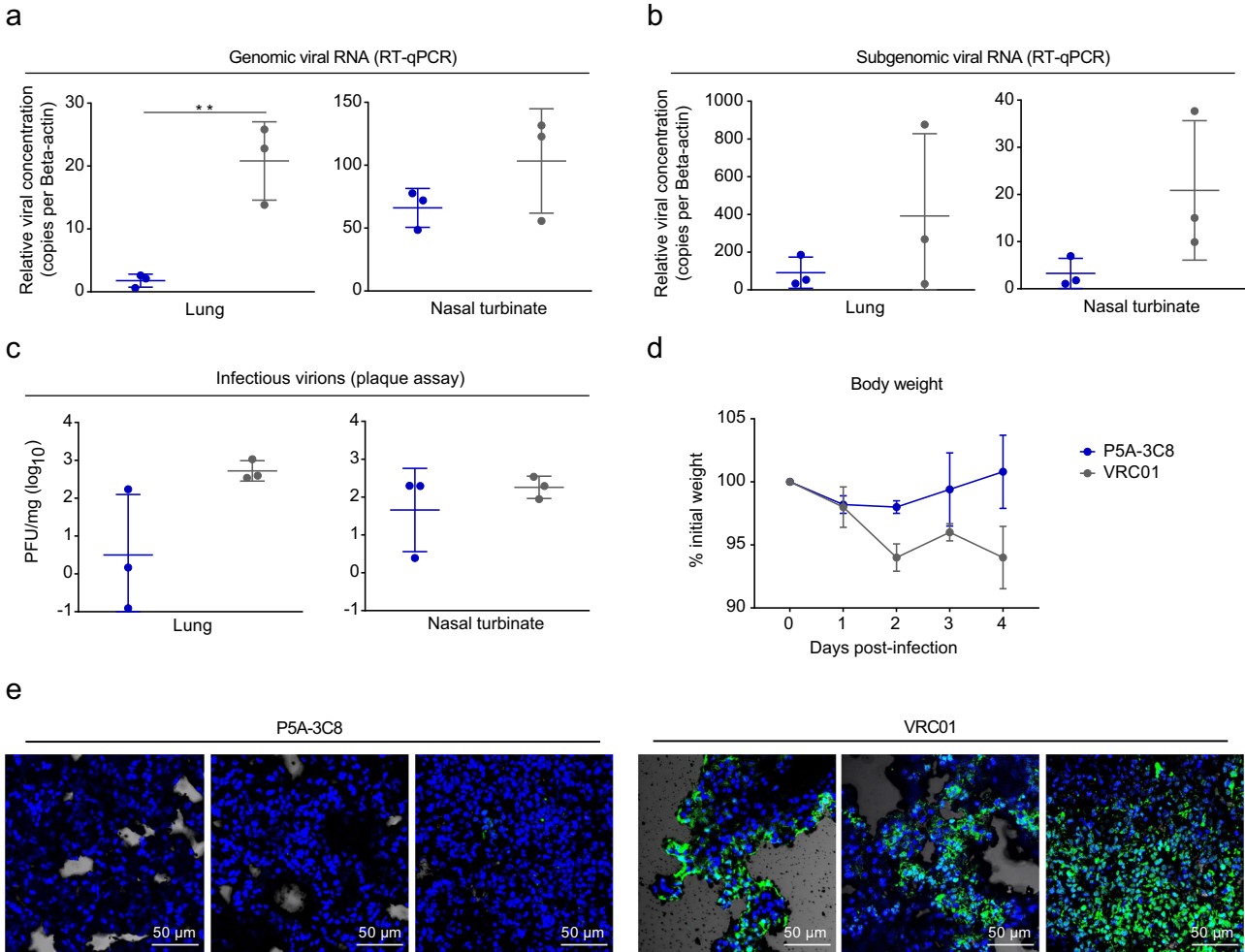

**Fig. 5 Efficacy of P5A-3C8 prophylaxis against live SARS-CoV-2 infection in Syrian hamsters. a** The hamsters were given a single intraperitoneal dose of 5 mg/kg of P5A-3C8 ($n = 3$), or VRC01, an anti-HIV-1 antibody as negative control ($n = 3$). On day 4 after viral challenge, the genomic viral RNA in the lung and nasal turbinate tissues were determined by qRT-PCR normalized by beta-actin. The differences between P5A-3C8 group and VRC01 group in lung tissues are statistically significant with two-tailed $p$ value = 0.0064 (**$p < 0.01$, unpaired $t$ test). **b** Subgenomic viral RNA in the lung and nasal turbinate tissues on day 4 after viral challenge were determined by qRT-PCR normalized by beta-actin ($n = 3$). **c** Infectious virions were tested by viral plaque assay in lung and nasal turbinate tissues. PFUs per mg of tissue extractions were compared between two groups ($n = 3$). **d** The body weights of hamsters were monitored over a 4-day time course ($n = 3$). All data from **a**–**d** are shown in mean value ± SD. **e** Representative images of hamster lung tissues detected for viral NP antigen by immunofluorescence. In the VRC01 group, diffuse NP expression was shown in large areas of alveoli. Sporadic NP expression were observed in lung sections of hamster treated with P5A-3C8. All images were magnified ×200. Scale bar: 50 μm. Results presented in **e** are representatives of two independent experiments. Source data (**a**–**d**) are provided as a Source Data file.

identified K417N/T mutations conferring resistance to this class of antibodies, like those in class I or RBS-A indicated by previous studies[44,45]. Three out of four public antibodies lost their activity due to this mutation while one remains sensitive. Especially, K417N/T has already been identified in SA501Y.V2 and BR501Y. V3 mutant strains, decreasing or abolishing neutralizing activity of mAbs including those (CB6 and REGN10933) already approved for Emergency-Use-Administration, convalescent plasma from naturally infected patients, and immune sera from vaccinated individuals[28,36–40,46,47]. Next generation vaccines to induce antibodies targeting more evolutionarily conserved site other than receptor-binding site (RBS) would be of great importance. Further, K417N mutation has also been identified in mouse-adapted SARS-CoV-2 strains together with N501Y and Q493H[48]. These combined mutations significantly enhanced the binding to mouse ACE2 and improve replication capacity during adaption process in mice, raising more serious concerns about this particular mutation[48]. Thus, the identification and characterization of public antibodies and their escape variants help us to better understand the interplay between antibody response and viral evolution, and to assist us to design next generation antibody combination and vaccines to overcome SARS-CoV-2 variants.

## Methods

**Antibody and Fab fragment production.** The production of antibodies was conducted as previously described[49]. The genes encoding the heavy and light chains of isolated antibodies were separately cloned into expression vectors containing IgG1 constant regions and the vectors were transiently transfected into HEK 293F cells (ATCC) using polyethylenimine (PEI) (Sigma). After 96 h, the antibodies secreted into the supernatant were collected and captured by protein A Magnetic beads (Genscript). The bound antibodies were eluted and further purified by gel-filtration chromatography using a Superdex 200 High Performance column (GE Healthcare). To produce Fab fragments, antibodies were cleaved using Protease Lys-C (Sigma) with an IgG to Lys-C ratio of 4000:1 (w/w) in 10 mM EDTA, 100 mM Tris-HCl, pH 8.5 at 37 °C for ~12 h. Fc fragments were removed using Protein A Sepharose (GE Healthcare).

**Neutralization activity of mAbs against wild typed and mutant pseudovirus SARS-CoV-2.** Single mutations were introduced to SARS-CoV-2 gene, with QuickChange Site-directed mutagenesis Kit (Agilent 210518) followed the manufacturer's instructions. SARS-CoV-2 were generated by co-transfection of human immunodeficiency virus backbones expressing firefly luciferase (pNL43R-E-luciferase) and pcDNA3.1 (Invitrogen) expression vectors encoding the SARS-Cov-2 S proteins or mutants into 293T cells (ATCC). Viral supernatants were collected 48 h later. Viral titers were measured as luciferase activity in relative light units (Bright-Glo Luciferase Assay Vector System, Promega Biosciences). Neutralization assays were performed by incubating pseudoviruses with serial dilutions of purified mAbs at 37 °C for 1 h. Huh7 cells (~$1.5 \times 10^4$ per well) (ATCC) were added in duplicate to the virus-antibody mixture. Half-maximal inhibitory concentrations ($IC_{50}$) of the evaluated mAbs were determined by luciferase activity 48 h after exposure to virus-antibody mixture using GraphPad Prism 7 (GraphPad Software Inc.).

**Neutralization activity of mAbs against live SARS-CoV-2.** SARS-CoV-2 focus reduction neutralization test (FRNT) was performed in a certified Biosafety level 3 laboratory. Neutralization assays against live SARS-CoV-2 were conducted using a clinical isolate (Beta/Shenzhen/SZTH-003/2020, EPI_ISL_406594 at the Global Initiative on Sharing All Influenza Data (GISAID) database) previously obtained from a nasopharyngeal swab of an infected patient. Serial dilutions of testing antibodies were conducted, mixed with 50 µL of SARS-CoV-2 (100 focus-forming unit) in 96-well microwell plates and incubated at 37 °C for 1 h. Mixtures were then transferred to 96-well plates seeded with Vero E6 cells (ATCC) and allowed absorption for 1 h at 37 °C. Inoculums were then removed before adding the overlay media (100 µL minimal essential media containing 1.6% carboxymethylcellulose). The plates were then incubated at 37 °C for 24 h. Overlays were removed and then cells were fixed with 4% paraformaldehyde solution for 30 min, permeabilized with Perm/Wash buffer (BD Biosciences) containing 0.1% Triton X-100 for 10 min. Cells were incubated with rabbit anti-SARS-CoV-2-N antibody (Sino Biological 40588-T62, 1:1000 dilution) for 1 h at room temperature before adding Horseradish Peroxidase (HRP)-conjugated goat anti-rabbit IgG (Heavy chain + Light chain) antibody (TransGen Biotech HS101-01, 1:2000 dilution). The reactions were developed with TrueBlue Peroxidase substrates (Seracare Life Sciences Inc 5510-0030). The numbers of SARS-CoV-2 foci were calculated using an EliSpot reader (Cellular Technology Ltd).

**Gene family usage and alignment analysis of mAbs.** The program IMGT/V-QUEST (http://www.imgt.org/IMGT_vquest/vquest) was used to analyze germline gene, germline divergence or degree of SHM, the framework region (FR) and the loop length of the complementarity determining region 3 (CDR3) for each antibody clone. The IgG heavy and light chain variable genes were aligned using Clustal W in the BioEdit sequence analysis package (https://bioedit.software.informer.com/7.2/).

**Recombinant RBDs and receptor ACE2.** Recombinant RBDs and the N-terminal peptidase domian of human ACE2 were expressed using the Bac-to-Bac baculovirus system (Invitrogen) as previously described[49,50]. SARS-CoV-2 RBD (residues Arg319-Lys529) and ACE2 (residues Ser19-Asp615) containing the gp67 secretion signal peptide and a C-terminal $His_6$ tag was inserted into pFastBac-Dual vectors (Invitrogen) and transformed into DH10 Bac component cells (ThermoFisher). The bacmid was extracted and further transfected into $Sf9$ cells (ATCC) using Cellfectin II Reagents (Invitrogen). The recombinant viruses were harvested from the transfected supernatant and amplified to generate high-titer virus stock. Viruses were then used to infect $Sf9$ cells for RBD production. Secreted RBD and ACE2 were harvested from the supernatant and purified by gel-filtration chromatography.

**Antibody binding kinetics and competition with receptor ACE2 measured by SPR.** The binding kinetics and affinity of mAbs to SARS-CoV-2 RBD were analyzed by SPR (Biacore T200, GE Healthcare). Specifically, protein A (Sino Biological) was firstly covalently immobilized onto a CM5 sensor chip, followed by capture of the individual antibodies and then injection of purified soluble SARS-CoV-2 WT and K417R/A/E/N/T mutant RBDs at five different concentrations. SPR assays were run at a flow rate of 30 µL/min in PBS buffer (with 0.05% Tween-20). The sensograms were fit in a 1:1 binding model with BIA Evaluation software (GE Healthcare). To determine competition with the human ACE2 peptidase domain, SARS-CoV-2 RBD was immobilized to a CM5 sensor chip via amine group for a final RU around 250. Antibodies (1 µM) were injected onto the chip until binding steady-state was reached. ACE2 (2 µM) was then injected for 60 s. Blocking efficacy was determined by comparison of response units with and without prior antibody incubation.

**Crystallization and data collection.** The SARS-CoV-2 RBD and the Fab fragment of P5A-1D2, P5A-3C8, and P22A-1D1 were respectively mixed at a molar ratio of 1:1.2, incubated for 2 h at 4 °C and further purified by gel-filtration chromatography. The purified complex concentrated to ~10 mg/mL in HBS buffer (10 mM HEPES, pH 7.2, 150 mM NaCl) was used for crystallization. The screening trials were performed at 18 °C using the sitting-drop vapor-diffusion method by mixing

0.2 µL of protein with 0.2 µL of reservoir solution. Crystals were successfully obtained in 0.2 M Magnesium chloride hexahydrate, 0.1 M Tris, pH 8.5, 3.4 M 1,6-Hexanediol for P5A-1D2, 0.2 M Lithium sulfate monohydrate, 0.1 M HEPES, pH 7.5, 25% w/v PEG 3350 for P5A-3C8, and 0.1 M potassium chloride, 0.1 M HEPES, pH 7.0, 15% PEG 5000MME for P22A-1D1, respectively. Diffraction data were at 100 K and at a wavelength of 0.97918 Å on the BL17U1 beam line of the Shanghai Synchrotron Research Facility. Diffraction data were auto-processed with aquarium pipeline and the data processing statistics are listed in Supplementary Table 2.

**Structure determination and refinement.** The structure was determined by the molecular replacement method with PHASER (CCP4 Interface 7.1.007)[51]. The search models were the SARS-CoV-2 RBD structure (PDB ID: 6M0J) and the structures of the variable domain of the heavy and light chains available in the PDB with the highest sequence identities (5I1E for VH model; 4LRN for VL model; 2XTJ for CH model; 6U3Z for CL model). Subsequent model building and refinement were performed using COOT v.0.9.2 and PHENIX v.1.18.2, respectively[52,53]. Final Ramachandran statistics: 95% favored, 3.9% allowed and 0.81% outliers for the final RBD-P22A-1D1 complex structure; Final Ramachandran statistics: 94.23% favored, 5.44% allowed, and 0.32% outliers for the final RBD-P5A-1D2 complex structure; Final Ramachandran statistics: 97% favored, 3.1% allowed and 0.33% outliers for the final RBD-P5A-3C8 complex structure.

**Analysis of antibody binding to cell surface expressed wild-type and mutant spike protein.** Single mutations were conducted with QuickChange Site-directed mutagenesis Kit (Agilent 210518) followed the manufacturer's instructions. HEK 293T cells were transfected with expression plasmid encoding either wild-type or mutant full-length SARS-CoV-2 and incubated at 37 °C for 36 h. The cells were removed from the plate using trypsin and distributed into 96-well plates for the individual staining. Cells were kept at 4 °C or on ice in the following incubation or wash steps. Cells were washed twice with 200 µL ice-cold staining buffer (PBS with 2% heated-inactivated FBS) between each of the following. The cells were stained for 1 h in 100 µL staining buffer with 10 µg/mL ACE2 protein or 2 µg/mL mAbs. The cells were then stained with one of the following secondary antibodies: anti-his PE (Miltenyi Biotec 130-120-787, 1:200 dilution) for ACE2, anti-human IgG Fc PE (Biolegend 410708, 1:40 dilution) for nAbs, or anti-mouse IgG FTIC (Thermo-Fisher A16073, 1:100 dilution) for S2 mAb (MP 08720401, 1:200 dilution). Finally, the cells were re-suspended and analyzed with FACS Calibur instrument (BD Biosciences, USA) and FlowJo 10 software (FlowJo, USA). HEK 293T cells without mock transfection were stained as background control.

**Generation of replication-competent recombinant VSVs.** The rVSV vector was constructed as described previously[54]. Briefly, the eGFP encoding sequences were added at nt62 to generate the rVSV-eGFP-G plasmid and the humanized spike protein coding sequence of SARS-CoV-2 Wuhan-Hu-1 strain (GeneBank: YP_009724390.1) was inserted into rVSV-eGFP-G plasmid to replace VSV glycoprotein coding sequence (3845-5380), generating rVSV-eGFP-SARS-CoV-2 plasmid. To rescue the rVSVs, HEK293T cells were transfected with rVSV-eGFP-SARS-CoV-2 plasmid and five supporting plasmids encoding T7 polymerase, N, P, M, and L of VSV using the calcium phosphate method. Virus in the supernatant were harvested and passaged on Vero cells to obtain virus stocks.

**Antibody escape studies.** Antibodies were serially diluted 1: 5 starting at 50 µg/mL in 100 µl of rVSV media and a total of $4 \times 10^5$ focus-forming units per ml (FFU/ml) of rVSV-eGFP-SARS-CoV-2 virus in 100 µL of medium was added to each dilution and incubate at 37 °C for 1 h. Then the mixture was added to $1 \times 10^5$ Vero cells and incubated for 4 days at 37 °C, 5% $CO_2$. Virus replication was monitored by screening for GFP positive cells using Opera Phenix (Perkin Elmer). The supernatants and cellular layers were collected from wells with evident viral replication. The total RNA, including the viral RNA, was extracted from the cells using TRIzol (Life Technologies) and reverse-transcribed (SuperScript III Reverse Transcriptase, Invitrogen, 18080-044) for subsequent sequencing of the SARS-CoV-2 spike gene. For a second round of selection, 100 µL of supernatant containing the virus was passed to antibodies starting at 250 µg/mL in 100 µL of rVSV media. Again, the supernatants and cellular layers were collected with the lowest antibody concentration and evident viral replication. The total RNA was extracted, transcribed and sent for sequencing of the SARS-CoV-2 spike gene, as above described. The sequences of primers are shown in Supplementary Table 6.

**Hamster protection studies.** Protection efficacy of public antibody P5A-3C8 was evaluated in an established golden hamster model of SARS-CoV-2 infection as described previously with slight changes[55]. Approval was obtained from the University of Hong Kong (HKU) Committee on the Use of Live Animals in Teaching and Research (CULATR5518-20). 6–8-week-old male and female hamsters were provided from the Chinese University of Hong Kong Laboratory Animal Service Centre through the HKU Laboratory Animal Unit and kept in Biosafety Level-2 housing with access to standard pellet feed and water ad libitum until virus challenge in the BSL-3 animal facility. Each hamster ($n = 3$ per group) was intraperitoneally administered one dose of 5 mg/kg of public antibody P5A-3C8 or

VRC01 as control. One day later, each hamster was intranasally challenged with 100 μL of Dulbecoo's Modified Eagle Medium containing $10^3$ PFU of SARS-CoV-2 (HK13 strain, GenBank accession no: MT835140) under intraperitoneal ketamine (200 mg/kg) and xylazine (10 mg/kg) anesthesia. Each group of hamsters were monitored twice daily for clinical signs of disease and sacrificed 4 days after virus challenge. Lung and nasal turbinate tissues were harvested for viral load determined by quantitative SARS-CoV-2-specific RdRp/Hel reverse transcription-polymerase chain reaction assay[56], detection of the subgenomic viral RNA containing the N gene utilizing specific primers (sgmRNA-F-44 and sgmRNA-R-N-28458)[57], and infectious virus titration by plaque assay[55,58]. The viral load for genomic RNA and subgenomic RNA was normalized by beta-actin while the PFU was normalized by mg of tissue extractions based on hamster's bodyweight. The sequences of primers are shown in Supplementary Table 6.

**Histopathology and immunohistochemistry**. IF staining was conducted on deparaffinized and rehydrated lung tissue sections for identification and localization of SARS-CoV-2 nucleocapsid protein (NP), using rabbit anti-SARS-CoV-2-N antibody together with Alexa Fluor 488-conjugated goat anti-rabbit IgG antibody. The lung tissues were treated with antigen unmasking solution in pressure cooker (Vector Laboratories), and blocked with 0.1% Sudan black B for 15 min and 1% bovine serum albumin BSA/PBS at RT for 30 min. This was followed by incubation with primary antibody rabbit anti-SARS-CoV-2-N antibody (Sino Biological 40588-T62, 1:1000 dilution) at 4 °C overnight and Alexa Fluor 488-conjugated goat anti-rabbit IgG antibody (Thermo Fisher Scientific A11034, 1:1000 dilution) for 30 min, and then mounted with 4′,6-diamidino-2-phenylindole (DAPI). The images of the lung tissues were captured with Olympus BX53 semi-motorized fluorescence microscope using cellSens imaging software.

**Reporting summary**. Further information on research design is available in the Nature Research Reporting Summary linked to this article.

## Data availability

Data generated or analyzed during this study are included in this published article (and its supplementary information files). All other data are also available from the corresponding author upon reasonable requests. The coordinates and structure factor files for the P5A-1D2, P5A-3C8, and P22A-1D1/SARS-CoV-2 RBD complexes have been deposited in the Protein Data Bank (PDB) under accession numbers 7CHO (https://www.rcsb.org/structure/7CHO), 7CHP (https://www.rcsb.org/structure/7CHP), 7CHS (https://www.rcsb.org/structure/7CHS), respectively. The sequences of P5A-1D2, P5A-3C8, and P22A-1D1 antibodies have been deposited in GenBank with accession codes MZ151910 (P5A-1D2HC), MZ151911 (P5A-1D2LC), MZ151912 (P5A-3C8HC), MZ151913 (P5A-3C8KC), MZ151914 (P22A-1D1HC), and MZ151915 (P22A-1D1KC). Source data are provided with this paper.

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

## Acknowledgements

We acknowledge the work and contribution of all the health providers from Shenzhen Third People's Hospital. We also thank patients for their active participation. We thank Dr. Aihua Zheng from Institute of Zoology at Chinese Academy of Science for providing recombinant VSV strain. This study was supported by the National Science Fund for Distinguished Young Scholars (82025022), the National Key Plan for Scientific Research and Development of China (2020YFC0848800, 2020YFC0844200, and 2016YFD0500307), the Science and Technology Innovation Committee of Shenzhen Municipality (202002073000002, 2020A1111350032, JCYJ20190809115617365, and JCYJ20180228162229889), the Shenzhen Science and Technology Program (RCYX20200714114700046), the Shenzhen Bay Fund (2020B1111340074), Bill & Melinda Gates Foundation and by Tsinghua University Initiative Scientific Research Program (20201080053) and Tsinghua University Spring Breeze Fund (2020Z99CFG004).

This work is also partially supported by the National Natural Science Foundation Award (81530065, 91442127, and 82002140), Beijing Municipal Science and Technology Commission (171100000517-001 and -003), Guangdong Basic and Applied Basic Research Foundation (2021B1515020034, 2019A1515011197, 2020B1111340074), Beijing Advanced Innovation Center for Structural Biology at Tsinghua University, Tencent Foundation, Shuidi Foundation, and TH Capital. The funders had no role in study design, data collection, data analysis, data interpretation, or writing of the report.

## Author contributions

Z.Z., X.W., L.Z., Z.C., and Y.W. conceived and designed the study. Q.Z., B.J., and J.G. performed most of the experiments together with assistance from J.C., L.C., R.W., W.H., P.C., M.F., B.Z., Shuo Song, Sisi Shan, Li Liu, B.Y., S.Z., X.G., J.Y., Q.L., L.F., and X.S. Q.Z. and L.C. performed and analyzed pseudo typed and live SARS-CoV-2 neutralization assay. J.Y., J.G., and X.W. provided assistance in RBD and trimeric spike protein production. J.G. with assistance from R.W. solved and analyzed crystal structure of antibody and RBD complex. M.F. conducted the virus escape experiment. J.C. and Li Liu performed the animal study. Lei Liu played critical roles in recruitment and clinical management of the study subjects. H.W. and J.Z. are in charge of sample collection and processing. Q.Z., B.J., J.G., J.C., L.C., R.W., W.H., Y.W., Z.C., L.Z., X.W., and Z.Z. had full access to data in the study, generated figures and tables, and take responsibility for the integrity and accuracy of the data presentation. L.Z., X.W., and Z.Z. wrote the manuscript. All authors reviewed and approved the final version of the manuscript.

## Competing interests

The authors declare the following competing interests: Patent applications have been filed on monoclonal antibodies targeting SARS-CoV-2 (patent application number: PCT/CN2020/108718; patent applicants: The Third People's Hospital of Shenzhen and Tsinghua University). Q.Z., B.J., X.S. Lei Liu, L.Z., and Z.Z. are the inventors. All other authors declare no competing interests.

## Additional information

[1]NexVac Research Center, Comprehensive AIDS Research Center, Beijing Advanced Innovation Center for Structural Biology, School of Medicine, Tsinghua University, Beijing, China. [2]Institute for Hepatology, National Clinical Research Center for Infectious Disease, Shenzhen Third People's Hospital; The Second Affiliated Hospital, School of Medicine, Southern University of Science and Technology, Shenzhen, Guangdong Province, China. [3]The Ministry of Education Key Laboratory of Protein Science, Beijing Advanced Innovation Center for Structural Biology, Beijing Frontier Research Center for Biological Structure, Collaborative Innovation Center for Biotherapy, School of Life Sciences, Tsinghua University, Beijing, China. [4]State Key Laboratory of Emerging Infectious Diseases, The University of Hong Kong, Pokfulam, Hong Kong SAR, People's Republic of China. [5]Department of Microbiology, Li Ka Shing Faculty of Medicine, The University of Hong Kong, Pokfulam, Hong Kong SAR, People's Republic of China. [6]Division of HIV/AIDS and Sex-Transmitted Virus Vaccines, National Institutes for Food and Drug Control,

Beijing, China. [7]Shenzhen Bay Laboratory, Shenzhen, Guangdong Province, China. [8]AIDS Institute, Li Ka Shing Faculty of Medicine, The University of Hong Kong, Pokfulam, Hong Kong SAR, People's Republic of China. [9]Institute of Biopharmaceutical and Health Engineering, Tsinghua Shenzhen International Graduate School, Tsinghua University, Shenzhen, China. [10]Institute of Biomedical Health Technology and Engineering, Shenzhen Bay Laboratory, Shenzhen, China. [11]These authors contributed equally: Qi Zhang, Bin Ju, Jiwan Ge, Jasper Fuk-Woo Chan, Lin Cheng, Ruoke Wang, Weijin Huang. ✉email: wangyc@nifdc.org.cn; zchenai@hku.hk; zhanglinqi@tsinghua.edu.cn; xinquanwang@mail.tsinghua.edu.cn; zhangzheng1975@aliyun.com

