## [Peer Review File · Nature Communications]

REVIEWER COMMENTS

Reviewer #1 (Remarks to the Author):

In this article, Zhang and colleagues studied antibodies found that a large proportion of neutralising antibodies (nAbs) isolated from a pool of eight patients derived from heavy chain germlines V(H)3-53 and V(H)3-66. All these nAbs competes with hACE2, neutralise rVSV bearing coronavirus spikes, and some of them protects syriam hamsters challenged with SARS-CoV-2. Detailed epitope mapping using X-ray crystallography showed that a representative number of these antibodies bind the same epitope in the RBD. Interestingly, they found using rVSV-S that a single mutation, K417N, is enough to abrogate binding to many of these antibodies, and allows viral escape.

I found that the manuscript is well written, but the authors should make an additional effort to improve some figures. In particular, I found that Figures 2 and 3, which show the structural analysis of the epitopes and paratopes of several nAbs, are not clear. In figure 2B the outline of the binding site of ACE2 is in green and the surface of the RBD in cyan, which generates very little contrast. The panels B and C in figure 3 can also be much clearer. Leaving aside the aesthetic aspects, I find the first part of the article (the structural description of the public antibodies) not original. The structural characterization of antibodies derived from germ line V(H)3-53 is reported in Science since some months (10.1126/science.abd2321) and the authors should include a succinct comparative analysis with these structures. However, I find really interesting the mutation K417N. The authors mention that this mutation circulates naturally. I would say even more, this mutation is one of those present in the so-called South African variant that is spreading in Europe. I think the authors should dedicate a paragraph to this variant, comparing it with the WT sequence, and including affinity data for hACE2. The authors should also discuss, based on their structural data, why some antibodies seem to be sensitive to this variant and others are not, when Y33 is present in all of them. Why K417A only reduces binding but K417N or K417T blocks it completely?, Which other mutations in this position would have the same effect ? Eventually, the authors should include a biophysical study (SPR) of different mutations. The authors should discuss the implications of their findings in the vaccination strategies. Are other mutations in K417 reported in the large genome database different than K417N?, Is K417T present? Another question that comes to mind, do the authors know if there are antibodies with the same gene usage that are non-neutralizing ?

I also have some minor comments:

1. Line 100, replace angel by angle
2. The fitting curve in Figure S1C for P5A-2G7 is weird with a large jump between the stages of association and dissociation. The same effect, although to a lesser extent can be observed for other

antibodies. This suggests a problem with blank subtraction and can lead to incorrect affinity values. Can authors reprocess the data to avoid these artifacts ?

3. For SARS-CoV-2 RBD-P5A-1D2 complex crystal. The resolution range for data collection (50-2.6) is narrower than for structure refinement (36-2.56), which of course is not possible. Can the authors correct these values ?

4. Which are the "ligands" in the crystallographic table ?

Reviewer #2 (Remarks to the Author):

In this manuscript 147 neutralizing monoclonal antibodies isolated from 8 donors against the SARS-CoV-2 receptor binding domain are evaluated for their ability to neutralize SARS-CoV-2 pseudovirus. 13 of these were found to have neutralizing potency (IC50) in the 1-100ng/ml range. These mAbs also neutralized authentic SARS-CoV-2. 7 of the 13 mAbs were derived from VH3-53/3-66 class of mAbs that have been described by several other groups. Most were from one donor #5 while 1 was isolated from donor #22 and 2 from #2. The crystal structures of 3 mAbs were solved in complex with the RBD and found to mimic the same mode of interaction of other antibodies of this class, although some subtle differences were noted. Alanine scanning mutagenesis of contact residues on the RBD revealed that Y421A and F456A affected binding of all the VH3-53/3-66 class mAbs tested, while the others had differential effects on mAb binding. One of the antibodies P4A-3C8 was tested in a syrian hamster infection model. A 5mg/kg dose of P5A-3C8 resulted in a reduction of lung viral load titers compared to an isotype control in response to a 1000 PFU intranasal challenge.

Replication competent VSV pseud typed with SARS-CoV-2 S was grown in the presence of two VH3-53/3-66 mAbs. Resistant strains were recovered and found to contain either a K417T or K417N mutation. Pseudoviruses with these mutations confirmed that they were resistant to the mAbs used to select for these mutations.

Overall, the study is clearly laid out and agrees with several published studies which have shown nearly identical data eg: Structural convergence from multiple donors (Yuan et al Science 369, Issue 6507, pp. 1119-1123) and protection in a challenge model (Rogers et al Science Vol. 369, Issue 6506, pp. 956-963).

The finding that this class of mAbs could select for the K417 mutation in the VSV assay is novel as far as I can tell. This mutation is found in several emerging viral variants that are now of considerable concern. I would recommend updating the text to highlight this. The authors cite the frequency of this mutation as of November 2020, I suspect it is more prevalent right now.

Could the authors comment, from a structural perspective why an alanine mutation at 417 can still bind P5AC38, but the mAb can't neutralize either K417T or K417N

Reviewer #3 (Remarks to the Author):

The manuscript by Zhang and colleagues describes the structural characterization of neutralizing antibodies (nAbs) against the SARS-CoV-2 spike protein, in vivo efficacy testing of a single nAbs in the hamster model and analysis of escape mutants in vitro. The authors discuss the importance of escape mutants for nAbs treatment against SARS-CoV-2.

Strength: The work is timely and important. The structural analysis and in vitro seems well done.

Weakness: The in vivo work is weak and preliminary and could be expanded.

Major Comments

(1) The group numbers (n=3) for the in vivo testing of a single nAb in the hamster model is low and unlikely to provide power. Aside of virus replication in the lung, the impact on virus shedding would have been important to assess. This could have been easily done with nasal washes or oral swabs. Infectivity titers were not determined which would be more helpful than genome copies. At the very minimum genomic and subgenomic PCRs should be performed to assess viral replication more reliably.

(2) Why were not all four nAbs tested in the hamster model? Additionally, combination therapy would have been helpful to assess virus escape mutant development in vivo.

(3) Regarding escape mutant development, did the authors sequence isolated virus from the lung tissue or any other hamster tissue? This would provide data on virus escape in vivo and strengthen the work.

Minor Comments:

(1) The manuscript would benefit from a language check.

(2) Line 65: "no available treatments or vaccines" needs to be updated and corrected as treatment and vaccine options are being used, partially even licensed.

Point to Point Responses

Reviewer #1 (Remarks to the Author):

1. In this article, Zhang and colleagues studied antibodies found that a large proportion of neutralising antibodies (nAbs) isolated from a pool of eight patients derived from heavy chain germlines V(H)3-53 and V(H)3-66. All these nAbs competes with hACE2, neutralise rVSV bearing coronavirus spikes, and some of them protects syriam hamsters challenged with SARS-CoV-2. Detailed epitope mapping using X-ray crystallography showed that a representative number of these antibodies bind the same epitope in the RBD. Interestingly, they found using rVSV-S that a single mutation, K417N, is enough to abrogate binding to many of these antibodies, and allows viral escape.

I found that the manuscript is well written, but the authors should make an additional effort to improve some figures. In particular, I found that Figures 2 and 3, which show the structural analysis of the epitopes and paratopes of several nAbs, are not clear.

In figure 2B the outline of the binding site of ACE2 is in green and the surface of the RBD in cyan, which generates very little contrast. The panels B and C in figure 3 can also be much clearer.

Response: According to reviewer's suggestion, we have changed the outline of the binding site of ACE2 from green to black in Fig. 2 as indicated below and revised Fig. 3B and 3C.

Figure 2. Structure and binding features of public antibodies to SARS-CoV-2 RBD.

Figure 3. Shared binding interface among the public antibodies.

2. Leaving aside the aesthetic aspects, I find the first part of the article (the structural description of the public antibodies) not original. The structural characterization of antibodies derived from germ line V(H)3-53 is reported in Science since some months (10.1126/science.abd2321) and the authors should include a succinct comparative analysis with these structures.

Response: Thank you for your suggestions. In the revised manuscript, we have compared the binding surface area of IGHV3-53/3-66 antibodies with those from other studies, such as B38 (Wu et al., 2020, Science), CC12.1 and CC12.3 (Yuan et al., 2020, Science), CV30 (Hurlburt et al., 2020, Nature Communications). We found that the light chains of P5A-3C8 and P22A-1D1 (in our study) and B38 and CC12.1 (in other's study) use IGKV1-9, which engages a larger binding interface than those of IGKV3-20 used by P2C-1F11, CC12.3 and CV30 (Table S3).

Table S3. Comparison of buried surface area of light chain IGKV1-9 and IGKV3-20 usage among public antibodies.

	P5A-1D2		P5A-3C8		P22A-1D1		P2C-1F11		B38		CB6		CC12.1		CC12.3		C105		CV30	
	H	L	H	L	H	L	H	L	H	L	H	L	H	L	H	L	H	L	H	L
CDR3	IGHV 3-53	IGLV 1-40	IGHV 3-53	IGKV 1-9	IGHV 3-53	IGKV 1-9	IGHV 3-66	IGKV 3-20	IGHV 3-53	IGKV 1-9	IGHV 3-66	IGKV 1-39	IGHV 3-53	IGKV 1-9	IGHV 3-53	IGKV 3-20	IGHV 3-53	IGLV 2-8	IGHV 3-53	IGKV 3-20
Buried surface (Å ²)	839.6	164.0	725.2	547.9	728.7	408.8	774.8	204.3	736.3	486	732.6	355.4	786.3	560.2	724.0	167.0	677.7	266.2	791.5	247.6
Total (Å ²)	1003.6		1273.1		1137.5		979.1		1222.3		1088		1346.5		891		943.9		1039.1	

3. However, I find really interesting the mutation K417N. The authors mention that this mutation circulates naturally. I would say even more, this mutation is one of those present in the so-called South African variant that is spreading in Europe. I think the authors should dedicate a paragraph to this variant, comparing it with the WT sequence, and including affinity data for hACE2.

Response: This is a great suggestion. In the revised manuscript, we have described that K417N/T has already been identified in SA501Y.V2 and BR501Y.V3 mutant strains, decreasing or abolishing neutralizing activity of monoclonal antibodies including that (CB6 and REGN10933) already approved for EUA, convalescent plasma from naturally infected patients, and immune sera from vaccinated individuals¹⁻⁸. We also

conducted additional experiment to test the binding affinity of ACE2 to all of the occurring K417 mutant RBDs (K417R/E/N/T) and K417A (Response Fig #1). Compared to WT RBD, K417R mutant RBD increased binding by ACE2 about 3.3-fold, while other mutants had negligible impact. Similar biochemical properties in charge and side chain between K and R may explain such results.

Response Fig #1. Binding affinity of K417R/A/E/N/T RBD mutants to ACE2.

4. The authors should also discuss, based on their structural data, why some antibodies seem to be sensitive to this variant and others are not, when Y33 is present in all of them. Why K417A only reduces binding but K417N or K417T blocks it completely? Which other mutations in this position would have the same effect? Eventually, the authors should include a biophysical study (SPR) of different mutations.

Response: According to the reviewer's suggestion, we have utilized SPR to test the binding affinity of the public antibodies to all the occurring K417 mutant (K417R/E/N/T) and K417A RBDs. As shown in Fig. 4, public antibodies bind K417R RBD in a similar level as WT, except that P2C-1F11 has 4.4-fold increase. In contrast, K417A/E/N/T mutants have profound impact on binding by P22A-1D1, P5A-1D2 and P5A-3C8 (Fig. 4).

In addition, we also compared neutralizing activities of the public antibodies against pseudovirus bearing WT, K417R, K417A, K417E, K417N, or K417T mutant SARS-CoV-2 spike on the pseudovirus. P22A-1D1, P5A-3C8 and P2C-1F11 remained sensitive to K417R pseudovirus, while P5A-1D2 had neutralizing activity below the detection limit (BDL) even when tested at the highest concentration (1 μ g/mL) (Fig. 4). K417A/E/N/T pseudoviruses were fully resistant to the public antibodies except for P2C-1F11 (Fig. 4).

We have also conducted detailed analysis on the interactions between the public antibodies and SARS-CoV-2 K417 residue within 4Å cutoff (Table S5). Salt bridges are highly involved in the binding between K417 residue and P22A-1D1, P5A-1D2 and P5A-3C8, but not with P2C-1F11. As such, K417A/E/N/T mutations are more disruptive to binding by P22A-1D1, P5A-1D2 and P5A-3C8, leading to loss of neutralizing activity. In addition, K417N mutation, together with N501Y and Q493H, has also been identified in mouse-adapted SARS-CoV-2 strains⁹. These combined mutations significantly enhanced the binding affinity to mouse ACE2 and improved replication capacity during adaptation in mice.

Figure 4. Susceptibility of SARS-CoV-2 K417 variants to binding and neutralization of public antibodies. Values indicate the fold changes in (A) binding affinity (K_D) and (B) half-maximal inhibitory concentrations (IC_{50}). The symbol “-”

indicates increased resistance and “+” increased sensitivity. Those K_D or IC_{50} values highlighted in red, resistance increased at least two-fold; in blue, sensitivity increased at least two-fold; and in white, resistance or sensitivity increased less than two-fold. BDL (Below Detection Limit) indicates the highest concentration of mAbs failed to bind or reach 50% neutralization. (C) The individual antibodies were captured on protein A covalently immobilized onto a CM5 sensor chip followed by injection of purified soluble SARS-CoV-2 WT and K417/R/A/E/N/T mutant RBDs at five different concentrations. The black lines indicate the experimentally derived curves while the red lines represent fitted curves based on the experimental data. (D) Comparison of public antibodies’ neutralization against pseudovirus bearing WT, K417R, K417A, K417E, K417N, or K417T mutant SARS-CoV-2 spike on the pseudovirus. VRC01 is an HIV-1 specific antibody used here as a negative control. Data shown were from at least two independent experiments.

Table S5. Interactions between public antibodies and SARS-CoV-2 K417 residue.

	RBD	antibody	Length (Å)	Interaction	Total
P2C-1F11	E/K417/N[N]	H/Y52/CE2[C]	3.84		7
		H/Y52/OH[O]	3.87	Hydrogen bond	
	E/K417/CG[C]	H/Y52/CE2[C]	3.85		
		H/Y52/CZ[C]	3.97		
		H/Y52/OH[O]	3.32		
	E/K417/CE[C]	H/Y52/OH[O]	3.69		
	E/K417/NZ[N]	H/Y52/OH[O]	3.19	Hydrogen bond	
P22A-1D1	E/K417/N[N]	H/Y52/CE2[C]	3.76		11
	E/K417/CG[C]	H/Y52/CE2[C]	3.65		
		H/Y33/OH[O]	3.53		
		H/Y52/OH[O]	3.87		
	E/K417/CD[C]	H/Y33/OH[O]	3.48		
		H/D100/OD2[O]	3.85		
	E/K417/CE[C]	H/D100/OD1[O]	3.94		
		H/D100/OD2[O]	3.71		
	E/K417/NZ[N]	H/D100/CG[C]	3.72		
		H/D100/OD1[O]	3.83	Salt bridge	
		H/D100/OD2[O]	2.87	Salt bridge	
P5A-1D2	E/K417/CG[C]	H/Y33/OH[O]	3.74		13
		H/Y52/CE2[C]	3.65		
	E/K417/CD[C]	H/Q100/OE1[O]	3.57		
	E/K417/CE[C]	H/Q100/OE1[O]	3.59		
		H/D106/OD1[O]	3.74		
		H/Y52/OH[O]	3.76		
	E/K417/NZ[N]	H/Q100/CB[C]	3.99		
		H/Q100/CG[C]	3.83		
		H/Q100/CD[C]	3.55		
		H/Q100/OE1[O]	2.61	Hydrogen bond	
		H/D106/CG[C]	3.32		
		H/D106/OD1[O]	2.70	Salt bridge	
		H/D106/OD2[O]	3.21	Salt bridge	
P5A-3C8	E/K417/N[N]	H/Y52/CE2[C]	3.73		11
	E/K417/CG[C]	H/Y33/OH[O]	3.18		
		H/Y52/CE2[C]	3.90		
	E/K417/CD[C]	H/Y33/OH[O]	3.66		
		H/Q100/OE1[O]	3.29		
	E/K417/CE[C]	H/Q100/OE1[O]	3.59		
	E/K417/NZ[N]	H/E101/OE2[O]	3.96	Salt bridge	
		H/E101/OE1[O]	3.99	Salt bridge	
		H/Q100/CD[C]	3.90		
		H/Q100/OE1[O]	2.82	Hydrogen bond	
	L/L91/CD2[C]	3.78			

5. The authors should discuss the implications of their findings in the vaccination strategies.

Response: Thank you for your suggestion. We have added the implications of our novel findings for vaccine strategies in the last paragraph of the discussion part in the revised manuscript.

6. Are other mutations in K417 reported in the large genome database different than K417N? Is K417T present?

Response: Yes. K417T has already emerged in BR501Y.V3 mutant strains (Brazil). We analyzed the viral sequences of a total of 592, 944 in the GISAID database (by March 16th, 2021), and found 1,022 sequences bearing the K417N mutation, 491 sequences containing K417T mutation, 4 sequences with K417R mutation, and 2 sequences with K417E mutation (<https://cov.lanl.gov/content/index>¹⁰)

7. Another question that comes to mind, do the authors know if there are antibodies with the same gene usage that are non-neutralizing ?

Response: Yes, we have identified four non-neutralizing IGVH3-53/3-66 antibodies out of total 15 (Response Table #1). Together with Dr. Wilson's finding in which NY motif and SGGS motif partially contribute to the binding of IGVH3-53/3-66 antibodies to the RBD¹¹, we propose that the neutralizing ability of a particular antibody is determined by multiple factors including actual sequence of HCDR3, heavy and light chain usage, the buried binding surface on the RBD and so on.

Response Table #1. Neutralizing activity, and gene family analysis of 15 IGVH3-53/3-66 antibodies isolated from Patient #1, Patient #2, Patient #5, and Patient #22.

mAbs	Pseudovirus (µg/ml)		Heavy chain							Light chain						
	IC ₅₀	IC ₄₀	V	J	D	SHM (%)	CDR1	CDR2	CDR3	CDR3 length	V	J	SHM (%)	CDR3	CDR3 length	
P22A-1D1	0.0098	0.0623	3-53*01	6*02	-	0.00	GFTVSSNY	IYSGGST	ARDRDVYGMVDV	11	KC	1-9*01	1*01	0.38	LHLNSVRT	8
P5A-1D1	0.0098	0.0691	3-53*01	6*02	3-16*01	0.35	GLTVSSNY	IYSGGST	ARDLYYYGMVDV	11	KC	1-9*01	5*01	0.76	QLNSYPT	8
P5A-1B8	0.0113	0.0991	3-53*01	4*02	2-15*01	1.40	GFTVSSNY	IYSGGST	ARETLAFDY	9	KC	1-9*01	4*01	0.00	QLNSYPPA	9
P5A-3C8	0.0286	0.1031	3-53*01	6*02	4-11*01	1.05	GFTVSSNY	IYSGGST	ARDLQEHGMVDV	11	KC	1-9*01	2*01	1.14	QHLNSYPGYTT	11
P2C-1F11	0.0286	0.1195	3-66*01,3-66*04	6*02	2-15*01	1.75	GFTVSSNY	IYSGGST	ARDLVVYGMVDV	11	KC	3-20*01	2*01,2*02	0.00	QYQSSPT	8
P5A-3A1	0.9231	4.2357	3-53*01	4*02	4-17*01	0.00	GFTVSSNY	IYSGGST	ARDYGDYFEDY	11	KC	3-20*01	2*02	0.00	QYQSSPRT	9
P2B-1G1	4.2200	11.6200	3-66*01,3-66*04	5*02	4-17*01	0.00	GFTVSSNY	IYSGGST	ARDYGDYWFDP	11	KC	3-20*01	2*02	0.00	QYQSSPRT	9
P5A-1D2	0.0186	0.1025	3-53*01	4*02	1-26*01	1.40	GFTVSSNY	IYSGGST	ARALQVGAITSDFDY	15	LC	1-40*01	2*01,3*01	1.11	QSCDSSLVVVV	11
P2B-1A10	0.0929	0.7446	3-53*01	3*02	1-20*01	0.35	GFTVSSNY	IYSGGST	AREGPKSTIGTAFDI	15	KC	1-33*01,1D-33*01	2*01	0.38	QYDNLPMYT	10
P2C-1D7	0.2100	1.6700	3-53*01	4*02	1-26*01	0.00	GFTVSSNY	IYSGGST	ARELYEVGATDY	12	KC	2D-30*01	3*01	0.00	MQRYYTLAGV	9
P2B-1F5	10.3000	>50	3-53*01	4*02	2-2*01	0.00	GFTVSSNY	IYSGGST	ARALPAAGYFEDY	14	KC	1-NL1*01	1*01	0.00	QYVSTPPT	9
P1A-1D1	>50	>50	3-53*01	4*02	6-13*01	4.21	GFTVSSNY	IYSGGST	ARRVNTSWAHAS	12	LC	2-8*01	1*01	2.22	GSYGGSNFV	10
P1A-1D5	>50	>50	3-53*01	6*02	2-15*01	1.05	GFTVSSNY	IYSGGST	ARVGLPRYYYYGMVDV	15	KC	1-33*01,1D-33*01	3*01	0.00	QHYDNLVLT	9
P1A-1D6	>50	>50	3-53*01	6*02	2-15*01	4.56	GFTVSSNY	IYSGGST	ARVGLPRYYYYGMVDV	15	KC	1-33*01,1D-33*01	3*01	3.79	QYDILLPT	9
P2C-1E1	>50	>50	3-66*01,3-66*04	4*02	5-12*01	0.00	GFTVSSNY	IYSGGST	VYSGYVDY	9	KC	3-11*01	1*01	0.00	QQRSNWPSGT	10

Minor comments:

1. Line 100, replace angel by angle

Response: We have changed it in the revised manuscript.

2. The fitting curve in Figure S1C for P5A-2G7 is weird with a large jump between the stages of association and dissociation. The same effect, although to a lesser extent can be observed for other antibodies. This suggests a problem with blank subtraction and can lead to incorrect affinity values. Can authors reprocess the data to avoid these artifacts?

Response: Thank you for your corrections. We have repeated the experiment with individual antibodies captured on protein A covalently immobilized CM5 sensor chip, followed by injection of purified soluble SARS-CoV-2 WT RBD at five different concentrations. The black lines indicate the experimentally derived curves while the red lines represent fitted curves based on the experimental data.

3. For SARS-CoV-2 RBD-P5A-1D2 complex crystal. The resolution range for data collection (50-2.6) is narrower than for structure refinement (36-2.56), which of course is not possible. Can the authors correct these values ?

Response: We have corrected them in the revised manuscript (Table S2).

Supplementary Table 2. Data collection and refinement statistics (molecular replacement).

Data collection			
	SARS-CoV-2 RBD-P22A-1D1 complex	SARS-CoV-2 RBD-P5A-1D2 complex	SARS-CoV-2 RBD-P5A-3C8 complex
Wavelength (Å)	0.97918	0.97918	0.97918
Resolution range (Å)	50.00-2.40 (2.46-2.40) *	50.00-2.56 (2.66-2.56) *	68.22-2.36 (2.48-2.36) *
Space group	C2	C2	P 2 ₁ 2 ₁ 2
Unit cell dimensions			
a , b , c (Å)	193.34, 86.60, 57.16	158.75, 67.51, 154.37	112.54, 171.57, 54.87
α , β , γ (°)	90, 99.25, 90	90, 112.18, 90	90, 90, 90
Unique reflections	36392 (3573)	47159 (3117)	43632 (4424)
Completeness (%)	100.0 (100.0)	95.9 (64.3)	97.3 (100.0)
Mean I/sigma (I)	13.1 (1.9)	9.5 (1.0)	13.0 (2.4)
Redundancy	6.7 (7.0)	6.3 (3.7)	13.2 (13.8)
R _{merge} (%)	11.4 (98.3)	11.7 (79.9)	13.7 (115.7)
R _{pim} (%)	6.2 (42.7)	7.0 (41.2)	5.6 (46.4)
CC1/2	0.989 (0.655)	0.989 (0.450)	0.997 (0.808)
Wilson B-factor (Å ²)	37.35	51.17	45.42
Structure refinement			
Resolution (Å)	47.71-2.40	36.52-2.56	50.98-2.36
R _{work} /R _{free} (%)	17.7/18.4	20.3/24.7	19.4/21.9
No. atoms			
Protein	4797	9446	4770
Glycan	14	28	14
Protein residues	625	1249	623
B-factors (Å ²)			
Protein	45.63	53.90	47.25
Glycan	76.80	115.29	52.72
RMSD			
Bonds length (Å)	0.008	0.008	0.010
Bonds angles (°)	1.06	1.07	1.25
Ramachandran plot			
Favored (%)	95	94.23	97
Allowed (%)	3.9	5.44	3.1
Outliers (%)	0.81	0.32	0.33

4. Which are the "ligands" in the crystallographic table ?

Response: Ligands in the crystallographic table refer to NAG.

Reviewer #2 (Remarks to the Author):

1. In this manuscript 147 neutralizing monoclonal antibodies isolated from 8 donors against the SARS-CoV-2 receptor binding domain are evaluated for their ability to neutralize SARS-CoV-2 pseudovirus. 13 of these were found to have neutralizing potency (IC₅₀) in the 1-100ng/ml range. These mAbs also neutralized authentic SARS-CoV-2. 7 of the 13 mAbs were derived from VH3-53/3-66 class of mAbs that have been described by several other groups. Most were from one donor #5 while 1 was isolated from donor #22 and 2 from #2. The crystal structures of 3 mAbs were solved in complex with the RBD and found to mimic the same mode of interaction of other antibodies of this class, although some subtle differences were noted. Alanine scanning mutagenesis of contact residues on the RBD revealed that Y421A and F456A affected binding of all the VH3-53/3-66 class mAbs tested, while the others had differential effects on mAb binding. One of the antibodies P4A-3C8 was tested in a syrian hamster infection model. A 5mg/kg dose of P5A-3C8 resulted in a reduction of lung viral load titers compared to an isotype control in response to a 1000 PFU intranasal challenge. Replication competent VSV pseudotyped with SARS-CoV-2 S was grown in the presence of two VH3-53/3-66 mAbs. Resistant strains were recovered and found to contain either a K417T or K417N mutation. Pseudoviruses with these mutations confirmed that they were resistant to the mAbs used to select for these mutations.

Overall, the study is clearly laid out and agrees with several published studies which have shown nearly identical data eg: Structural convergence from multiple donors (Yuan et al Science 369, Issue 6507, pp. 1119-1123) and protection in a challenge model (Rogers et al Science Vol. 369, Issue 6506, pp. 956-963). The finding that this class of mAbs could select for the K417 mutation in the VSV assay is novel as far as I can tell. This mutation is found in several emerging viral variants that are now of considerable concern. I would recommend updating the text to highlight this. The authors cite the frequency of this mutation as of November 2020, I suspect it is more prevalent right now.

Response: Thank you for your comments and suggestions. In the revised manuscript, we have described that K417N/T has already been identified in SA501Y.V2 and

BR501Y.V3 mutant strains, decreasing or abolishing neutralizing activity of monoclonal antibodies including that (CB6 and REGN10933) already approved for EUA, convalescent plasma from naturally infected patients, and immune sera from vaccinated individuals¹⁻⁸. We have updated the frequency of this mutation with data from March 16th, 2021.

2. Could the authors comment, from a structural perspective why an alanine mutation at 417 can still bind P5AC38, but the mAb can't neutralize either K417T or K417N.

Response: We have utilized SPR to test the binding affinity of the public antibodies to all the occurring K417 mutant (K417R/E/N/T) and K417A RBDs and neutralization activities against pseudoviruses bearing WT and various K417 mutant SARS-CoV-2 spike (Fig. 4 in the revision). Compared to WT, the binding affinity of P5A-3C8 decreased 564.8-fold to K417A, 15.9-fold to K417E, 36.9-fold to K417N, 57.1-fold to K417T, resulting in loss of neutralizing activity against all these four mutant pseudovirus (Fig. 4). However, P5A-3C8 could still bind to K417R RBD and neutralize K417R pseudovirus with similar potency to WT. We found that salt bridges were highly involved in the binding between K417 residue and P5A-3C8. K417A/E/N/T mutations would be expected to disrupt these salt bridges, leading to loss of neutralizing activity. By contrast, K417R maintained the positive-charged side chain, resulting in minimal effect. We have listed the interactions between public antibodies and SARS-CoV-2 K417 residue within 4Å cutoff in Table S5 of the revision.

Reviewer #3 (Remarks to the Author):

Major Comments

1. The group numbers (n=3) for the in vivo testing of a single nAb in the hamster model is low and unlikely to provide power.

Response: Thank you for your suggestion. Due to limited animal resource, we could not perform additional experiments for now but will validate our results when more animals become available.

2. Aside of virus replication in the lung, the impact on virus shedding would have been important to assess. This could have been easily done with nasal washes or oral swabs.

Response: Unfortunately, we didn't collect nasal wash or oral swab samples. Our collaborator, Dr. Zhiwei Chen from the University of Hong Kong, showed that potent neutralizing antibodies failed to completely inhibit viral shedding in the nasal cavity despite full suppression of viral replication using the Syrian hamster model¹². Specifically, Dr. Chen showed that antibodies ZDY20 (IC₅₀ 0.13 µg/ml and IC₉₀ 1.24 µg/ml, 10 mg/kg and 5 mg/kg), ZB8 (IC₅₀ 0.013 µg/ml and IC₉₀ 0.031 µg/ml, 4.5 mg/kg) and 2-15 (IC₅₀ 0.0007 µg/ml and IC₉₀ 0.04 µg/ml, 1.5 mg/kg) suppressed productive infection in lungs, but did not prevent robust SARS-CoV-2 infection in nasal turbinate of Syrian hamster. Since the amounts of ZDY20 and ZB8 detected in nasal wash was very low, the lack for sterile protection is probably due to poor distribution of neutralizing antibodies to nasal turbinate to outcompete the robust viral infection there.

3. Infectivity titers were not determined which would be more helpful than genome copies. At the very minimum genomic and subgenomic PCRs should be performed to assess viral replication more reliably.

Response: Thank you for your suggestion. We have added infectious virus titer data and subgenomic PCR data in Fig.5 in the revised manuscript. Compared to the VRC01 control group, the total viral loads in the P5A-3C8 group were significantly reduced in lung, demonstrated by both viral RNA (genomic RNA and subgenomic RNA) and infectious virus titer measured by plaque forming unit (PFU) (Fig. 5A, 5B, 5C).

Figure 5. Efficacy of P5A-3C8 prophylaxis against live SARS-CoV-2 infection in Syrian hamsters. (A) The hamsters were given a single intraperitoneal dose of 5 mg/kg of P5A-3C8 (n = 3), or VRC01, an anti-HIV-1 antibody as negative control (n = 3). On day 4 after viral challenge, the genomic viral RNA in the lung and nasal turbinate tissues were determined by qRT-PCR normalized by beta-actin. The differences between P5A-3C8 group and VRC01 group in lung tissues are statistically significant with $**p < 0.01$. (B) Subgenomic viral RNA in the lung and nasal turbinate tissues on day 4 after viral challenge were determined by qRT-PCR normalized by beta-actin. (C) Infectious virions were tested by viral plaque assay in lung and nasal turbinate tissues. PFUs per mg of tissue extractions were compared between two groups. (D) The body weights of hamsters were monitored over a 4-day time course. All data from A-D are shown in mean value \pm SD. (E) Representative images of hamster lung tissues detected for viral NP antigen by immunofluorescence. In the VRC01 group, diffuse NP expression was shown in large areas of alveoli. Sporadic NP expression were observed in lung sections of hamster treated with P5A-3C8. All images are magnified 200 \times .

4. Why were not all four nAbs tested in the hamster model? Additionally, combination therapy would have been helpful to assess virus escape mutant development in vivo.

Response: Due to limited animal resource, we were unable to test all four or combination of our antibodies.

5. Regarding escape mutant development, did the authors sequence isolated virus from the lung tissue or any other hamster tissue? This would provide data on virus escape in vivo and strengthen the work.

Response: It is a good suggestion. During the manuscript submission period, we have already started infectious viral titer assay and subgenomic qRT-PCR experiments, which used up all of the tissue homogenates. Unfortunately, we have not additional samples for virus gene sequencing.

Minor Comments:

1. The manuscript would benefit form a language check.

Response: We have checked carefully in the revised manuscript.

2. Line 65: “no available treatments or vaccines” needs to be updated and corrected as treatment and vaccine options are being used, partially even licensed.

Response: We have corrected them in the revised manuscript.

1 Andreano, E. *et al.* SARS-CoV-2 escape in vitro from a highly neutralizing COVID-19 convalescent plasma. *bioRxiv*, doi:10.1101/2020.12.28.424451 (2020).

2 FDA. *Coronavirus Disease 2019 (COVID-19) EUA Information*, <<https://www.fda.gov/emergency-preparedness-and-response/mcm-legal-regulatory-and-policy-framework/emergency-use-authorization#covid19euas>> (2021).

3 Greaney, A. J. *et al.* Complete Mapping of Mutations to the SARS-CoV-2 Spike Receptor-Binding Domain that Escape Antibody Recognition. *Cell host & microbe* **29**, 44-57.e49, doi:10.1016/j.chom.2020.11.007 (2021).

4 Huang, Y. *et al.* Identification of a conserved neutralizing epitope present on spike proteins from all highly pathogenic coronaviruses. *bioRxiv*, 2021.2001.2031.428824, doi:10.1101/2021.01.31.428824 (2021).

5 Wang, P. *et al.* Increased Resistance of SARS-CoV-2 Variants B.1.351 and B.1.1.7 to Antibody Neutralization. *bioRxiv*, doi:10.1101/2021.01.25.428137 (2021).

- 6 Wang, Z. *et al.* mRNA vaccine-elicited antibodies to SARS-CoV-2 and circulating variants. *Nature*, doi:10.1038/s41586-021-03324-6 (2021).
- 7 Weisblum, Y. *et al.* Escape from neutralizing antibodies by SARS-CoV-2 spike protein variants. *eLife* **9**, doi:10.7554/eLife.61312 (2020).
- 8 Wibmer, C. K. *et al.* SARS-CoV-2 501Y.V2 escapes neutralization by South African COVID-19 donor plasma. *Nature medicine*, doi:10.1038/s41591-021-01285-x (2021).
- 9 Sun, S. *et al.* Characterization and structural basis of a lethal mouse-adapted SARS-CoV-2. *bioRxiv*, 2020.2011.2010.377333, doi:10.1101/2020.11.10.377333 (2020).
- 10 Korber, B. *et al.* Tracking Changes in SARS-CoV-2 Spike: Evidence that D614G Increases Infectivity of the COVID-19 Virus. *Cell* **182**, 812-827.e819, doi:10.1016/j.cell.2020.06.043 (2020).
- 11 Yuan, M. *et al.* Structural basis of a shared antibody response to SARS-CoV-2. *Science (New York, N.Y.)* **369**, 1119-1123, doi:10.1126/science.abd2321 (2020).
- 12 Zhou, D. *et al.* Robust SARS-CoV-2 infection in nasal turbinates after treatment with systemic neutralizing antibodies. *Cell host & microbe*, doi:10.1016/j.chom.2021.02.019 (2021).

REVIEWERS' COMMENTS

Reviewer #1 (Remarks to the Author):

All my concerns have been addressed in the reviewed version. I would like to congratulate the authors for this excellent work.

Reviewer #2 (Remarks to the Author):

The authors have satisfied my concerns.

Reviewer #3 (Remarks to the Author):

The authors responded to my comments in a fair manner but were only partially able to strengthen the in vivo data.

In my view, the authors appropriately responded to the comments by the other reviewers.